# Global Convergence to Local Minmax Equilibrium in Classes of Nonconvex Zero-Sum Games

**Tanner Fiez**
University of Washington
fiezt@uw.edu

**Lillian J. Ratliff**
University of Washington
ratliffl@uw.edu

**Eric Mazumdar**
California Institute of Technology
mazumdar@caltech.edu

**Evan Faulkner**
University of Washington
evanjf5@uw.edu

**Adhyyan Narang**
University of Washington
adhyyan@uw.edu

## Abstract

We study gradient descent-ascent learning dynamics with timescale separation ($\tau$-`GDA`) in unconstrained continuous action zero-sum games where the minimizing player faces a nonconvex optimization problem and the maximizing player optimizes a Polyak-Łojasiewicz (PŁ) or strongly-concave (SC) objective. In contrast to past work on gradient-based learning in nonconvex-PŁ/SC zero-sum games, we assess convergence in relation to natural game-theoretic equilibria instead of only notions of stationarity. In pursuit of this goal, we prove that the only locally stable points of the $\tau$-`GDA` continuous-time limiting system correspond to strict local minmax equilibria in each class of games. For these classes of games, we exploit timescale separation to construct a potential function that when combined with the stability characterization and an asymptotic saddle avoidance result gives a global asymptotic almost-sure convergence guarantee for the discrete-time gradient descent-ascent update to a set of the strict local minmax equilibrium. Moreover, we provide convergence rates for the gradient descent-ascent dynamics with timescale separation to approximate stationary points.

## 1 Introduction

We study continuous action zero-sum games of the form

$$\min_{x \in \mathcal{X}} \max_{y \in \mathcal{Y}} f(x, y)$$

where $f \in C^2(\mathcal{X} \times \mathcal{Y}, \mathbb{R})$ and $\mathcal{X} = \mathbb{R}^{d_1}$ and $\mathcal{Y} = \mathbb{R}^{d_2}$ denote the individual action spaces and $d = d_1 + d_2$. In particular, we focus on unconstrained, continuous strategy space zero-sum games in which $f(\cdot, y)$ is potentially nonconvex in $x \in \mathcal{X}$ and $f(x, \cdot)$ satisfies the Polyak-Łojasiewicz (PŁ) condition [47] or is strongly-concave (SC) in $y \in \mathcal{Y}$. We refer to these classes of games as nonconvex-PŁ and nonconvex-SC zero-sum games, respectively.

This general formulation has a broad spectrum of applications such as fair classification [45], distributionally robust optimization [44, 48, 53], and adversarial training [38]. Consequently, there has been a surge of interest in recent years toward developing methods for solving these problems efficiently. So far, existing work on gradient-based learning in nonconvex-PŁ/SC zero-sum games has exclusively focused on providing global convergence rates to approximate stationary points with no attention given to the characterization in terms of game-theoretic equilibrium concepts [33, 34, 36, 45, 48, 62].

In contrast, a common theme in the study of general nonconvex-nonconcave zero-sum games is to assess the *types* of stationary points an algorithm locally converges toward in terms of their higher

35th Conference on Neural Information Processing Systems (NeurIPS 2021).

order structure [10, 15, 16, 23, 40, 41, 58, 64]. The purpose of this analysis is generally to determine whether commonly deployed algorithms can guarantee local convergence to *only* game-theoretic equilibria or to design algorithms that achieve this objective.

The goal of this paper is to close the gap between the two problem classes and determine whether gradient-based learning algorithms in nonconvex-PŁ/SC zero-sum games can be shown to globally converge to *only* game-theoretically meaningful equilibria (local minmax/Stackelberg or Nash). We focus our attention on the canonical gradient descent-ascent learning dynamics with timescale separation between players. In this algorithm, timescale separation is manifested in the different (yet constant) learning rates of the maximizing and minimizing players. The description of this system, which we refer to as $\tau$-GDA, where the ratio of learning rates denoted by $\tau > 0$ parameterizes the timescale separation between players, is provided in Algorithms 1–2 for deterministic and stochastic settings respectively. Simply put, $\tau$-GDA (Algorithm 1) corresponds to each player following their individual gradient in a noiseless setting, while stochastic $\tau$-GDA (Algorithm 2) describes each player following their individual stochastic gradient. The $\tau$-GDA update only requires first-order gradients and thus is a computationally efficient method for machine learning problems formulated as games.

## 1.1 Contributions

We show that $\tau$-GDA has *global* convergence guarantees in nonconvex-PŁ/SC zero-sum games to the natural game-theoretic solution concept for this problem class of strict local minmax equilibria.[1] The specific contributions of this work are now summarized.

1) In Theorem 1, we prove the only critical points that are locally stable with respect to the $\tau$-GDA continuous-time limiting system are strict local minmax equilibrium in nonconvex-PŁ/SC zero-sum games. A key implication of this is that any critical point which is not a strict local minmax equilibrium is a saddle point of the continuous-time limiting system.

2) In Theorem 2, we combine Theorem 1 with a potential function construction (Lemma 1) and an asymptotic saddle avoidance result (Lemma 2) to prove that the $\tau$-GDA update in deterministic settings (Algorithm 1) globally asymptotically converges to strict local minmax equilibria almost surely in the class of nonconvex-PŁ/SC zero-sum games.

3) In Corollary 1, using the potential function from Lemma 1, we show that $\tau$-GDA in deterministic settings reaches an $\varepsilon$-critical point in $\widetilde{\mathcal{O}}(\varepsilon^{-2})$ steps in nonconvex-PŁ/SC zero-sum games. Moreover, specific to nonconvex-SC zero-sum games, Lemma 3 shows there exists learning rates for $\tau$-GDA such that the cost function of the game itself can be made a potential. Corollary 2 then uses the potential function from Lemma 3 to show that $\tau$-GDA reaches an $\varepsilon$-critical point in $\widetilde{\mathcal{O}}(\varepsilon^{-2})$ and $\widetilde{\mathcal{O}}(\varepsilon^{-6})$ steps in deterministic and stochastic problems, respectively.

Prior to moving on, we briefly comment on and provide context for each contribution of this paper. As we discuss later on in Remark 1, beyond its importance toward proving Theorem 2, Theorem 1 is potentially of independent interest given the implications it also has for the local stability of $\tau$-GDA around critical points in the more general setting of nonconvex-nonconcave zero-sum games. Furthermore, to our knowledge, Theorem 2 provides the broadest existing global convergence guarantee for gradient-based algorithms to *game-theoretically meaningful* equilibria in zero-sum continuous games. Finally, while there exists known convergence rates for gradient-based learning algorithms to $\varepsilon$-critical points in nonconvex-PŁ/SC zero-sum games, we are unaware of such a result in nonconvex-PŁ zero-sum games for $\tau$-GDA, and the fact that we derive a rate in nonconvex-SC zero-sum games using the cost function itself as a potential function has practical implications given that it can easily be monitored when running the algorithm to evaluate progress toward a solution.

## 1.2 Practical Motivation

The study of nonconvex-PŁ/SC zero-sum games has often been motivated by machine learning problems formulated as games. We remark that given the problem formulations, it is natural from both game-theoretic and machine learning perspectives to seek notions of minmax equilibria. Indeed, notions of stationarity are not guaranteed to reflect a meaningful solution to the underlying problem. Consequently, it is critical to give convergence guarantees to minmax equilibrium as we pursue in

---

[1]Strict local minmax equilibria characterized by gradient-based sufficient conditions are also known as differential Stackelberg equilibria in the literature [15, 16] and we use the terms interchangeably in this work.

this work. We now provide examples from the literature of machine learning applications that are relevant to the class of games studied in this paper. Note that the following application problems are illustrative in nature and do not immediately fall into the classes of games we study. However, as is elaborated on shortly, each of the problems can be transformed into unconstrained nonconvex-PŁ/SC zero-sum games while retaining the optimization objectives after simple, standard modifications.

**Example 1.** In fair classification [45] and learning from multiple distributions, the objective is to minimize the maximum loss over multiple categories. An example formulation is the problem

$$\min_{w \in \mathbb{R}^d} \max_{i \in \{1, \ldots, K\}} \ell_i(w)$$

where $\ell_i(w)$ represents the loss on category $i$ with $w$ denoting neural network parameters. A reformulation of this problem [45] with $\mathcal{T}$ is the simplex in $\mathbb{R}^K$ is given by the zero-sum game

$$\min_{w \in \mathbb{R}^d} \max_{t \in \mathcal{T}} \sum_{i=1}^{N} t_i \ell_i(w). \tag{1}$$

**Example 2.** To train a neural network classifier robust against adversarial attacks, a common approach is to formulate training as a robust minmax optimization problem of the form

$$\min_{w \in \mathbb{R}^d} \sum_{i=1}^{N} \max_{\delta_i : |\delta_i|_\infty \leq \epsilon} \ell(x_i + \delta_i, y_i, w),$$

where $\ell(x_i + \delta_i, y_i, w)$ represents the loss on sample $x_i$ perturbed by $\delta_i$ with budget $\epsilon$ and label $y_i$ as a function of the parameter weights $w$ [38]. A reformulation of this problem [45] is given by

$$\min_{w \in \mathbb{R}^d} \sum_{i=1}^{N} \max_{t \in \mathcal{T}} \sum_{j=1}^{K} t_j \ell(\widehat{x}_{ij}, y_i, w) \tag{2}$$

where $\widehat{x}_{ij}$ is the result of a targeted attack on the sample $x_i$ seeking to change the output of the network to label $j$ and $\mathcal{T}$ is the simplex in $\mathbb{R}^K$.

**Example 3.** Distributionally robust optimization often results in a zero-sum game of the form

$$\min_{w \in \mathbb{R}^d} \max_{t \in \mathcal{T}} \sum_{i=1}^{N} t_i \ell_i(w) - r(t) \tag{3}$$

where $\ell_i(x)$ is the loss of a model $w$ on the $i$–th data point, $\mathcal{T}$ is the simplex in $\mathbb{R}^N$, and $r(t)$ is carefully selected convex regularizer [44, 48, 53].

The machine learning problem formulations in (1)–(3) from Examples 1–3 represent nonconvex-concave zero-sum games in which the strategy space of the minimizing player is unconstrained and the strategy space of the maximizing player is constrained to the simplex. These problems are naturally adapted to the unconstrained nonconvex-PŁ/SC zero-sum game setting considered in this paper by removing the constraint on the strategy space of the maximizing player and including a suitable PŁ/SC regularization penalty on the choice variable of the maximizing player.

## 1.3 Organization

The rest of the paper is organized as follows. Section 2 details related work and Section 3 presents game-theoretic, mathematical, and algorithmic preliminaries for our results. Section 4 is devoted to studying the local stability properties around critical points of the continuous-time limiting system for the $\tau$-GDA learning dynamics. In Section 5, we present our study of the convergence properties of $\tau$-GDA in nonconvex-PŁ/SC zero-sum games. We conclude with a discussion in Section 6. Finally, the supplementary material (appendix) contains the proofs of theoretical results.

## 2 Related Work

We now cover the most relevant related work with further discussion provided in Appendix A.

**Nonconvex-Nonconcave Zero-Sum Games.** A common theme in analyzing gradient descent-ascent with or without timescale separation in nonconvex-nonconcave zero-sum games has been to assess the local stability around critical points of the continuous-time limiting system and draw connections to local Nash and Stackelberg equilibrium notions characterized by gradient-based sufficient conditions (see Definition 4) [10, 15, 16, 23, 40, 42, 43, 63]. Importantly, it has been shown that unless the timescale separation is chosen very carefully, the stable critical points of gradient descent-ascent may not be game-theoretically meaningful and game-theoretically meaningful critical points may

Table 1: The gradient complexity of gradient descent (GD) and its perturbed variant (PGD), gradient descent-ascent with timescale separation ($\tau$-GDA), and alternating (including multi-step) gradient descent-ascent (AGDA) in deterministic and stochastic nonconvex optimization, nonconvex-SC zero-sum games, and nonconvex-PŁ zero-sum games. We state the complexity in terms of the $\varepsilon$ tolerance of the guarantee with the notation $\widetilde{\mathcal{O}}(\cdot)$ hiding logarithmic factors in $\varepsilon$.

| Problem | Algorithm & Reference | Complexity | | Guarantee |
|---|---|---|---|---|
| | | Deterministic | Stochastic | |
| Nonconvex | GD [24] | $\widetilde{\mathcal{O}}(\varepsilon^{-2})$ | $\widetilde{\mathcal{O}}(\varepsilon^{-4})$ | $\varepsilon$–Stationarity |
| Optimization | PGD [24] | $\widetilde{\mathcal{O}}(\varepsilon^{-2})$ | $\widetilde{\mathcal{O}}(\varepsilon^{-4})$ | $\varepsilon$–Local Min |
| Nonconvex | $\tau$-GDA, $\tau$-AGDA [33] | $\widetilde{\mathcal{O}}(\varepsilon^{-2})$ | $\widetilde{\mathcal{O}}(\varepsilon^{-5})$ | $\varepsilon$–Stationarity |
| SC Zero-Sum | **Theorem 2** ($\tau$-GDA) | Asymptotic | – | **DSE** |
| | **Corollary 1**, **Corollary 2** ($\tau$-GDA) | $\widetilde{\mathcal{O}}(\varepsilon^{-2})$ | $\widetilde{\mathcal{O}}(\varepsilon^{-6})$ | $\varepsilon$–Stationarity |
| Nonconvex | AGDA [45],[62, Appendix D] | $\widetilde{\mathcal{O}}(\varepsilon^{-2})$ | – | $\varepsilon$–Stationarity |
| PŁ Zero-Sum | **Theorem 2** ($\tau$-GDA) | Asymptotic | – | **DSE** |
| | **Corollary 1** ($\tau$-GDA) | $\widetilde{\mathcal{O}}(\varepsilon^{-2})$ | – | $\varepsilon$-Stationarity |

not be stable [15, 23]. We obtain stronger stability characterizations (see Theorem 1) for $\tau$-GDA in nonconvex-PŁ/SC zero-sum games using the structure imposed on the game cost function. The result of Theorem 1 also has novel implications for the local stability of $\tau$-GDA around critical points in the more general setting of nonconvex-nonconcave zero-sum games (see Remark 1). Moreover, it is fundamental toward proving the global almost sure asymptotic convergence guarantee for $\tau$-GDA in deterministic settings to strict local minmax equilibria in nonconvex-PŁ/SC zero-sum games (Theorem 2). Further discussion of this topic is given in Section 4.

**Nonconvex-PŁ and Nonconvex-SC Zero-Sum Games.** Table 1 provides a comprehensive comparison between our convergence results and existing convergence results for gradient descent variants in nonconvex optimization and nonconvex-PŁ/SC zero-sum games. We leave discussion of research on these classes of games with other algorithmic methods to Appendix A. The key distinction between this paper and past work is that instead of assessing convergence in terms of only reaching an approximate stationary point of the dynamics or a surrogate function, we obtain convergence guarantees in regards to differential Stackelberg equilibria (equivalently strict local minmax equilibria). Despite this being a much stricter and meaningful notion of solving the problem, Theorem 2 shows that global asymptotic convergence guarantees for $\tau$-GDA in deterministic settings to this type of solution remain obtainable in nonconvex-PŁ/SC zero-sum games. This is the main result of the paper. While we do not prove a rate of convergence to differential Stackelberg equilibria or an approximate notion, Corollary 1 shows $\tau$-GDA in deterministic settings with parameter choices satisfying the conditions of Theorem 2 reaches an $\varepsilon$-critical point in $\widetilde{\mathcal{O}}(\varepsilon^{-2})$ steps in nonconvex-PŁ/SC zero-sum games. We are unaware of an existing rate for $\tau$-GDA in deterministic settings in nonconvex-PŁ zero-sum games. Finally, as a complementary result, Lemma 3 shows there exists learning rates for $\tau$-GDA such that the cost function of the game itself can be made a potential in nonconvex-SC zero-sum games, which is a desirable property given that this potential function can be monitored while running the algorithm. Corollary 2 then uses this potential function to show that $\tau$-GDA reaches an $\varepsilon$-critical point in $\widetilde{\mathcal{O}}(\varepsilon^{-2})$ and $\widetilde{\mathcal{O}}(\varepsilon^{-6})$ steps in deterministic and stochastic nonconvex-SC zero-sum games, respectively.

**Escaping Saddle Points.** Saddle avoidance results for variants of gradient descent in nonconvex optimization are asymptotic or finite-time. The former states that almost surely the algorithm does not converge to saddles points [28, 29], while the latter gives rates of escape to conclude convergence to approximate local minimum [17, 20, 24]. A primary assumption in the aforementioned works is what is known as the strict saddle property, which ensures directions of escape exists from a saddle point. The methods for showing gradient descent escapes saddle points almost surely in nonconvex optimization have more recently been extended to gradient descent-ascent in the setting of continuous games under an analogous strict saddle assumption [10, 40]. A key component of proving the global asymptotic convergence guarantee to strict local minmax for $\tau$-GDA turns out to be ensuring that the

update escapes saddle points of the continuous-time limiting system almost surely. To show this property holds for $\tau$-GDA in nonconvex-PŁ/SC zero-sum games, we are able to invoke existing saddle avoidance results for gradient-based learning in continuous games [40].

## 3 Preliminaries

In the zero-sum games we study, we refer to the minimizing player controlling $x$ as player 1 and the maximizing player controlling $y$ as player 2. The set of players is denoted by $\mathcal{I} = \{1, 2\}$. We consider objective functions $f \in C^2(\mathcal{Z}, \mathbb{R})$ where the joint strategy space is denoted by $\mathcal{Z} = \mathcal{X} \times \mathcal{Y}$ where $\mathcal{X} = \mathbb{R}^{d_1}$ and $\mathcal{Y} = \mathbb{R}^{d_2}$ denote the individual action spaces and $d = d_1 + d_2$. We often denote a joint strategy using the shorthand notation $z = (x, y) \in \mathcal{Z} = \mathcal{X} \times \mathcal{Y}$.

**Notation.** We denote $\nabla f$ as the total derivative of $f$, $\nabla_i f$ as the derivative of $f$ with respect to the choice variable of player $i$, $\nabla_{ij} f$ as the partial derivative of $\nabla_i f$ with respect to the choice variable of player $j$, and $\nabla_i^2 f$ as the partial derivative of $\nabla_i f$ with respect to the choice variable of player $i$. We let $\| \cdot \|$ denote the 2-norm of vectors unless otherwise specified, $\mathrm{spec}(\cdot)$ denote the set of eigenvalues of a matrix, $\mathrm{Re}(\cdot)$ denote the real part of a complex number, and $\mathbb{C}_-^\circ$ and $\mathbb{C}_+^\circ$ denote the open left-half and right-half complex plane, respectively. Let $\lambda_{\min}(A)$ denote the eigenvalue of $A$ with the minimum real part, and $\lambda_{\max}(A)$ the eigenvalue of $A$ with the maximum real part. Finally, we indicate $A$ is positive and negative definite using the notation $A \succ 0$ and $A \prec 0$, respectively.

**Classes of Games.** We study and analyze both nonconvex-PŁ and nonconvex-SC zero-sum games. To begin, we state a standard smoothness assumption that is needed throughout.

**Assumption 1.** *Given a zero-sum game $(f, -f)$ defined by $f \in C^2(\mathcal{Z}, \mathbb{R})$, $\nabla_1 f(x, y)$ and $\nabla_2 f(x, y)$ are $L_1$ and $L_2$ Lipschitz, respectively. That is, $\forall\, x, x' \in \mathcal{X}, y, y' \in \mathcal{Y}$,*

$$\|\nabla_1 f(x, y) - \nabla_1 f(x', y')\| \le L_1(\|x - x'\| + \|y - y'\|),$$
$$\|\nabla_2 f(x, y) - \nabla_2 f(x', y')\| \le L_2(\|x - x'\| + \|y - y'\|).$$

This assumption immediately implies that the vector of individual gradients denoted by

$$g(x, y) := (\nabla_1 f(x, y), -\nabla_2 f(x, y))$$

is also Lipschitz with parameter $L = L_1 + L_2$.

We now define a nonconvex-PŁ zero-sum game [45]. This class of games allows for the objective to be nonconvex in $x \in \mathcal{X}$, but it needs to satisfy the Polyak-Łojasiewicz (PŁ) condition [47] in $y \in \mathcal{Y}$.

**Definition 1** (Nonconvex-PŁ Game). *Consider a zero-sum game $(f, -f)$ defined by $f \in C^2(\mathcal{Z}, \mathbb{R})$. The game is called nonconvex-PŁ if $f(x, \cdot)$ is $\mu$-PŁ with respect to the argument $y \in \mathcal{Y}$. That is, for $\mu > 0$ and for all $z = (x, y) \in \mathcal{Z}$,*

$$\|\nabla_2 f(x, y)\|^2 \ge 2\mu(\max_{y' \in \mathcal{Y}} f(x, y') - f(x, y)),$$

*where for any fixed $x \in \mathcal{X}$, $\max_{y' \in \mathcal{Y}} f(x, y')$ has a nonempty solution set and a finite optimal value.*

In nonconvex-PŁ zero-sum games, the solution set of $\mathrm{argmax}_{y' \in \mathcal{Y}} f(x^*, y')$ for any fixed $x^* \in \mathcal{X}$ may not be a singleton and the maximizing player's objective may be nonconcave. However, by Definition 1 it follows immediately that any $(x^*, y^*)$ such that $\|\nabla_2 f(x^*, y^*)\| = 0$ satisfies $f(x^*, y^*) = \max_{y' \in \mathcal{Y}} f(x^*, y')$ so any stationary point with respect to $y$ is a global maximum.

The following provides a definition for what we call a nonconvex-SC zero-sum game. In this class of games, the function that defines the game may be nonconvex in $x \in \mathcal{X}$, but it must be SC in $y \in \mathcal{Y}$.

**Definition 2** (Nonconvex-SC Game). *Consider a zero-sum game $(f, -f)$ defined by $f \in C^2(\mathcal{Z}, \mathbb{R})$. The game is nonconvex-SC if $f(x, \cdot)$ is $\mu$-SC with respect to the argument $y \in \mathcal{Y}$. That is, given any $x \in \mathcal{X}$ and for all $y, y' \in \mathcal{Y}$,*

$$f(x, y') \le f(x, y) + \langle \nabla_2 f(x, y), y' - y \rangle - \frac{\mu}{2} \|y' - y\|_2^2.$$

It is important to note that nonconvex, $\mu$-SC zero-sum games are nonconvex, $\mu$-PŁ zero-sum games, but nonconvex-PŁ zero-sum games may not be nonconvex-SC zero-sum games, which follows from the relationship between PŁ and SC functions [25]. This is to say that the class of nonconvex-PŁ

| **Algorithm 1** $\tau$-GDA | **Algorithm 2** Stochastic $\tau$-GDA |
|---|---|
| **Input:** $x_0 \in \mathbb{R}^{d_1}$, $y_0 \in \mathbb{R}^{d_2}$ | **Input:** $x_0 \in \mathbb{R}^{d_1}$, $y_0 \in \mathbb{R}^{d_2}$ |
| **for** $k = 0, 1, \ldots$ **do** | **for** $k = 0, 1, \ldots$ **do** |
| $\quad x_{k+1} \leftarrow x_k - \gamma \nabla_1 f(x_k, y_k)$ | $\quad x_{k+1} \leftarrow x_k - \gamma g_1(x_k, y_k; \theta_{1,k})$ |
| $\quad y_{k+1} \leftarrow y_k + \gamma \tau \nabla_2 f(x_k, y_k)$ | $\quad y_{k+1} \leftarrow y_k + \gamma \tau g_2(x_k, y_k; \theta_{2,k})$ |
| **end for** | **end for** |

zero-sum games is a superset of the class of nonconvex-SC zero-sum games. Thus, results we state for nonconvex-PŁ zero-sum games immediately hold for nonconvex-SC zero-sum games.

**Learning Dynamics.** We study gradient descent-ascent with timescale separation. Let $\gamma$ be the learning rate of player 1 and $\tau > 0$ be the timescale parameterization so that the learning rate of player 2 is given by $\tau\gamma$. The deterministic $\tau$-GDA dynamics are presented in Algorithm 1 and the stochastic $\tau$-GDA dynamics are given in Algorithm 2 (see Section 5.2 for the relevant notation). The key distinction between this study of gradient descent-ascent with timescale separation and past work is how we assess convergence as we now begin to formalize.

**Stationarity Notions.** We call *critical points* strategy profiles at which the individual gradient of each player is equal to zero; that is $(x, y) \in \mathcal{Z}$ such that $\nabla_1 f(x, y) = 0$ and $\nabla_2 f(x, y) = 0$. Critical points correspond to stationary points of the $\tau$-GDA dynamics. The nonconvex-PŁ/SC zero-sum game literature has generally assessed convergence in terms of the complexity of finding an $\varepsilon$-*critical point* (see, e.g., [45, 62]). We formally define an $\varepsilon$-critical point now for later reference.

**Definition 3** ($\varepsilon$-Critical Point.)**.** *The joint strategy* $(x^*, y^*) \in \mathcal{Z}$ *is an $\varepsilon$-critical point when the conditions* $\|\nabla_1 f(x^*, y^*)\| \leq \varepsilon$ *and* $\|\nabla_2 f(x^*, y^*)\| \leq \varepsilon$ *hold.*

A closely related and common notion of convergence in this body of work (see e.g., [33]) is that of finding an $\varepsilon$-critical point of the function $\max_{y \in \mathcal{Y}} f(\cdot, y)$. This criterion amounts to seeking to achieve the condition $\|\nabla_1 \max_{y \in \mathcal{Y}} f(x, y)\| \leq \varepsilon$.

In contrast, we assess convergence with connections to the equilibrium notions that are commonly studied in the nonconvex-nonconcave zero-sum game literature. Since either stationarity notion may lack any game-theoretic meaning, we consider a strictly harder notion of solving a game.

**Equilibrium Notions.** The typical solution concept in game theory when an implicit or explicit order of play is present in the structure of the game is the (local) Stackelberg (equivalently minmax in zero-sum games) equilibrium concept [6].[2] Informally, in nonconvex-PŁ/SC zero-sum games, a local minmax equilibrium corresponds to a strategy pair $(x^*, y^*) \in \mathcal{Z}$ such that $x^*$ is a local minimum of the function $f(x, y_*(x))$ where $y_*(x) \in \operatorname{argmax}_{y \in \mathcal{Y}} f(x, y)$ is a local maximum of the function $f(x^*, y)$. When the function $f$ is bounded or when $f(\cdot, y)$ is bounded and $f(x, \cdot)$ is strongly concave, a minmax equilibrium is guaranteed to exist.

We characterize the local minmax (Stackelberg) equilibrium notion in terms of sufficient conditions on player costs as is typical in learning in games (see, e.g., [15, 16, 23, 40, 58, 64]). Toward presenting this definition, we denote by $J(x, y)$ the Jacobian of the individual gradient vector $g(x, y)$ given by

$$J(x, y) = \begin{bmatrix} \nabla_1^2 f(x, y) & \nabla_{12} f(x, y) \\ -\nabla_{12}^\top f(x, y) & -\nabla_2^2 f(x, y) \end{bmatrix}. \tag{4}$$

Let $\mathtt{S}_1(\cdot)$ denote the Schur complement of $(\cdot)$ with respect to the $d_2 \times d_2$ block in $(\cdot)$. The following definition is characterized by sufficient conditions for a local minmax equilibrium in zero-sum games.

**Definition 4** (Differential Stackelberg/Strict Local Minmax Equilibrium [16, 23])**.** *The joint strategy* $(x^*, y^*) \in \mathcal{Z}$ *is a differential Stackelberg equilibrium when the conditions* $\nabla_1 f(x^*, y^*) = 0$, $\nabla_2 f(x^*, y^*) = 0$ *and* $\mathtt{S}_1(J(x^*, y^*)) \succ 0$, $\nabla_2^2 f(x^*, y^*) \prec 0$ *hold.*

We remark that in nonconvex-PŁ/SC zero-sum games, $\mathtt{S}_1(J(x^*, y^*))$ as in Definition 4 is well-defined at any critical point $(x^*, y^*)$ since $\det(\nabla_2^2 f(x^*, y^*)) \neq 0$ (see Lemma 8 in Appendix B.3).

---

[2]By implicit order of play, we mean the min and max order are not interchangeable.

# 4 Local Stability Analysis

To characterize the convergence of $\tau$-GDA, we begin by studying its continuous-time limiting system

$$\dot{z} = -\Lambda_\tau g(z) \tag{5}$$

where $\dot{z} = (\dot{x}, \dot{y})$, $\Lambda_\tau = \mathrm{blockdiag}(I_{d_1}, \tau I_{d_2})$, and $g(z)$ is the vector of individual gradients. The Jacobian of this system is given by

$$J_\tau(z) = \Lambda_\tau J(z). \tag{6}$$

We analyze the stability of the continuous-time system around critical points $z^* = (x^*, y^*)$ as a function of the timescale separation $\tau$ using the Jacobian $J_\tau(z^*)$ in this section toward drawing conclusions about the stability and convergence of the discrete time system $\tau$-GDA. A critical point is said to be locally (exponentially) stable when the spectrum of $-J_\tau(z^*)$ is in the open left-half complex plane $\mathbb{C}_-^\circ$ (cf. Theorem 3, Appendix B.2). Simply put, a critical point $z^*$ is locally exponentially stable if and only if the real parts of the eigenvalues of $-J_\tau(z^*)$ are strictly negative. Throughout, we use the broader term "stable" to mean the following.

**Definition 5** (Stability). *A critical point $z^* = (x^*, y^*) \in \mathcal{Z}$ is locally exponentially stable for $\dot{z} = -\Lambda_\tau g(z)$ if and only if* $\mathrm{spec}(-J_\tau(z^*)) \subset \mathbb{C}_-^\circ$ ($\equiv \mathrm{spec}(J_\tau(z^*)) \subset \mathbb{C}_+^\circ$).

Stability with respect to the continuous-time $\tau$-GDA dynamics guarantees that the system asymptotically converges at an exponential rate to the critical point in a local neighborhood. Moreover, given a suitable choice of learning rates, equivalent insights hold for the discrete-time dynamics [8].

**Stability in Nonconvex-Nonconcave Zero-Sum Games.** Given the implications regarding convergence, a number of papers in the past several years study the stability of $\tau$-GDA around critical points and the connections to differential Nash and Stackelberg equilibrium in zero-sum games [10, 15, 16, 23, 40]. However, this body of research focuses on general nonconvex-nonconcave zero-sum games. In general across the spectrum of nonconvex-nonconcave zero-sum games, the stable critical points of $\tau$-GDA coincide with the set of differential Stackelberg equilibria only when the timescale separation $\tau \to \infty$ [23]. Given that such a choice of timescale separation requires the learning rate $\gamma \to 0$ in order to retain stability of the discrete-time system, it is not clear how to derive a practical algorithm from this insight.

Toward remedying this problem, Fiez and Ratliff [15] provide stability results in terms of the timescale separation concerning a given critical point, rather than across the space of nonconvex-nonconcave zero-sum games. In particular, they prove a stability and instability result as a function of the timescale separation in the $\tau$-GDA dynamics. The stability results say that given a differential Stackelberg equilibrium $z^*$, there exists a finite $\tau^* \in (0, \infty)$ that can be constructed such that $z^*$ is stable for all $\tau \in (\tau^*, \infty)$. On the other hand, the instability results says that given a critical point which is not a differential Stackelberg equilibrium, there exists a finite $\tau_0 \in (0, \infty)$ that can be constructed such that $z^*$ is not stable for all $\tau \in (\tau_0, \infty)$.

**Stability in Nonconvex-PŁ/SC Zero-Sum Games.** To our knowledge, the connection between the stability (and instability) of critical points with respect to $\tau$-GDA dynamics and game-theoretic equilibrium notions in the semi-structured problems of nonconvex-PŁ and nonconvex-SC zero-sum games has not been fully characterized. We show in the following result that when a nonconvex-nonconcave game is specialized to a nonconvex-PŁ zero-sum game as from Definition 1, significantly more general stability characterizations can be obtained by exploiting the fact that $\nabla_2^2 f(x^*, y^*) \prec 0$ at any critical point $z^* = (x^*, y^*)$ (see Lemma 8 in Appendix B.3). Notably, any critical point $z^* = (x^*, y^*)$ that is not a differential Stackelberg equilibrium is not stable ("unstable") for all $\tau > 0$. In other words, $\tau$-GDA does not admit spurious stable points in nonconvex-PŁ zero-sum games. This is in stark contrast to the known fact that $1$-GDA admits spurious stable points in the more general class of nonconvex-nonconcave games as has been shown in previous literature [10, 23, 40]. Moreover, if $z^*$ is a differential Stackelberg equilibrium, then $z^*$ is stable for all $\tau$ larger than the minimum $\tau_*$ for which $z^*$ is stable and such a finite $\tau_*$ is guaranteed to exist. This result implies that in practice, one can select a finite value of $\tau$ to run $\tau$-GDA with, and all stable critical points (if they exist) will be differential Stackelberg equilibria, and if $\tau$ is scaled up and the set of stable points grows, then only differential Stackelberg equilibria can be introduced to the set.

**Theorem 1.** *Consider a nonconvex-PŁ/SC zero-sum game $(f, -f)$ where $f \in C^2(\mathcal{Z}, \mathbb{R})$. Then, the following hold: 1) Any critical point $z^* \in \mathcal{Z}$ that is not a differential Stackelberg equilibrium is unstable with respect to $\tau$-GDA for all $\tau \in (0, \infty)$; 2) If $z^*$ is a differential Stackelberg equilibrium,*

*then* $\operatorname{spec}(-J_\tau(z^*)) \subset \mathbb{C}_-^\circ$ *for all* $\tau \in [\tau_*, \infty)$ *where* $\tau_*$ *is the minimum* $\tau \in (0, \infty)$ *such that* $\operatorname{spec}(-J_\tau(z^*)) \subset \mathbb{C}_-^\circ$ *and a finite* $\tau_*$ *is guaranteed to exist.*

In comparison to the stability results for nonconvex-nonconcave zero-sum games of Fiez and Ratliff [15], we obtain the stronger results in nonconvex-PŁ zero-sum games that $(i)$ $\tau_0 = 0$ for any critical point that is not a differential Stackelberg equilibrium and $(ii)$ a differential Stackelberg equilibrium is never unstable after it becomes stable as a function of the timescale separation $\tau$.

**Remark 1.** The proof of Theorem 1 also reveals novel insights into the local stability of $\tau$-GDA around critical points in nonconvex-nonconcave zero-sum games. In particular, given any critical point $z^* = (x^*, y^*)$ in a nonconvex-nonconcave zero-sum game $(f, -f)$ where $f \in C^2(\mathcal{Z}, \mathbb{R})$ such that $\nabla_2^2 f(x^*, y^*) \prec 0$, if $z^*$ is not a differential Stackelberg equilibrium, then it is unstable with respect to $\tau$-GDA for all $\tau \in (0, \infty)$. This observation follows from the proof of Theorem 1 relying on the fact that $\nabla_2^2 f(x^*, y^*) \prec 0$ holds at any critical point in nonconvex-PŁ/SC zero-sum games.

# 5 Convergence Analysis

Theorem 1 in the previous section characterizes the local behavior of the continuous-time $\tau$-GDA dynamics around critical points. However, ultimately we are most interested in the convergence properties of the discrete-time $\tau$-GDA dynamics. This forms the focus of this section. Specifically, Section 5.1 is devoted to showing that $\tau$-GDA in deterministic settings almost surely asymptotically converges to strict local minmax equilibria in nonconvex-PŁ/SC zero-sum games and Section 5.2 develops convergence rates to $\varepsilon$-critical points in these classes of games.

## 5.1 Asymptotic Convergence to Strict Local Minmax Equilibrium

We prove in Theorem 2 of this subsection that the deterministic $\tau$-GDA (Algorithm 1) update almost surely converges to a differential Stackelberg equilibrium in nonconvex-PŁ/SC zero-sum games. Despite this result being asymptotic, to our knowledge it is the most general class of zero-sum games in which a global convergence guarantee to established game-theoretic equilibria has been given.

To begin, observe that the continuous-time $\tau$-GDA dynamics do not necessarily correspond to a gradient flow of a function, which can be observed from the fact that the Jacobian $J_\tau$ is not guaranteed to be symmetric. As a consequence, there may exist complex limiting behavior beyond convergence to a critical point such as non-trivial limit cycles or periodic orbits. To rule out this phenomenon, we prove that there exists a bounded potential function that decreases along the iterates of $\tau$-GDA and only stops decreasing at critical points of $\tau$-GDA. In what follows, $y_*(x)$ denotes an element of $\operatorname{argmax}_{y \in \mathcal{Y}} f(x, y)$ and we define $\kappa := L_1/\mu$. Moreover, as an implication of Assumption 1 and Definition 1, $\nabla \max_{y \in \mathcal{Y}} f(\cdot, y)$ is $L_4 = (L_1 + \kappa L_2)$ Lipschitz (see Lemma 7 in Appendix B.3).

**Lemma 1.** *Consider a nonconvex-PŁ/SC zero-sum game defined by* $f \in C^2(\mathcal{Z}, \mathbb{R})$ *satisfying Assumption 1. For any* $\Gamma \in (0, 1/7)$*, suppose that* $\tau \geq \Gamma^{-1} \kappa^2$ *and* $\gamma < \min\{\frac{1}{2L}, \frac{1}{2\tau L}, \frac{1}{2L_4}\}$*, then* $\Phi(x, y) = f(x, y_*(x)) - \Gamma f(x, y)$ *is a potential function for* $\tau$-GDA.

The function $f(x, y_*(x))$ can be seen as the function the $x$ player would minimize if the $y$ player was playing a best-response $y_*(x)$. The potential function $\Phi(x, y)$ essentially captures that along trajectories of $\tau$-GDA, either the value of $f(x, y_*(x))$ should decrease, or the value of $f(x, y_*(x)) - f(x, y)$ should decrease since the $y$-player converges at a fast rate to $y_*(x)$ given the time-scale separation. Indeed, this potential function implicitly guarantees that the maximizing player *tracks* the best response set, and that the minimizing player essentially ends up minimizing $f(x, y_*(x))$ as desired. The choice of $\tau$ and the learning rate $\gamma$ allow us to guarantee that this occurs.

Given the potential function in Lemma 1, we can conclude that Algorithm 1 converges to critical points. The critical points may correspond to stable points of the continuous-time $\tau$-GDA dynamics (Definition 5) or saddle points of the continuous-time $\tau$-GDA dynamics, which are defined as follows.

**Definition 6** (Saddle Point). *The critical point* $z^* = (x^*, y^*) \in \mathcal{Z}$ *is a saddle point of the* $\tau$-GDA *continuous-time dynamics* $\dot{z} = -\Lambda_\tau g(z)$ *if* $\operatorname{Re}(\lambda_{\max}(-J_\tau(z^*))) \geq 0$ *and a strict saddle if* $\operatorname{Re}(\lambda_{\max}(-J_\tau(z^*))) > 0$ *and* $\operatorname{Re}(\lambda) \neq 0 \, \forall \, \lambda \in \operatorname{spec}(-J_\tau(z^*))$.

Recall that Theorem 1 indicates the only stable points of the continuous-time $\tau$-GDA dynamics are differential Stackelberg equilibria. Thus, if we can show that the discrete-time $\tau$-GDA dynamics

avoid saddle points of the continuous-time $\tau$-GDA dynamics, then we can conclude that Algorithm 1 converges to only differential Stackelberg equilibria. The following result of Mazumdar et al. [40] states that the discrete-time $\tau$-GDA dynamics avoid strict saddle points of the continuous-time limiting system almost surely.[3]

**Lemma 2** (Theorem 4.1 [40]). *Consider a zero-sum game defined by the nonconvex-nonconcave function $f \in C^2(\mathcal{Z}, \mathbb{R})$ satisfying Assumption 1. The set of initial conditions $z \in \mathcal{Z}$ from which $\tau$-GDA with $\gamma < \min\{\frac{1}{L}, \frac{1}{\tau L}\}$ converges to strict saddle points of $\dot{z} = -\Lambda_\tau g(z)$ is of measure-zero.*

Lemma 2 can be combined with the existence of a potential function (Lemma 1) and the guarantee all stable points of the $\tau$-GDA continuous-time system are strict local minmax (Theorem 1) to arrive at a desirable global convergence guarantee for $\tau$-GDA when given the following assumption.

**Assumption 2** (Strict Saddle Property). *Given a zero-sum game $(f, -f)$ defined by $f \in C^2(\mathcal{Z}, \mathbb{R})$, we assume every saddle point of the continuous-time $\tau$-GDA dynamics given by the system $\dot{z} = -\Lambda_\tau g(z)$ from (5) is a strict saddle point for the choice of $\tau \in (0, \infty)$.*

Assumption 2 can be seen as an analogue for gradient dynamics in nonconvex zero-sum games to the commonly assumed strict saddle property in nonconvex optimization [17, 20, 24]. A key distinction is that in nonconvex optimization, strict saddles of the function and strict saddles of the gradient dynamics are equivalent, whereas this property does not carry over to the zero-sum game setting and instead the assumption needs to be explicitly made on saddle points of the gradient dynamics. Given this assumption, we are now ready to the state main result of this paper.

**Theorem 2.** *Consider a nonconvex-PŁ/SC zero-sum game $(f, -f)$ defined by $f \in C^2(\mathcal{Z}, \mathbb{R})$ that satisfies Assumptions 1–2. Then, $\tau$-GDA with $\tau \geq \Gamma^{-1}\kappa^2$, and $\gamma < \min\{\frac{1}{2L}, \frac{1}{2\tau L}, \frac{1}{2L_4}\}$ for $\Gamma \in (0, 1/7]$ asymptotically converges to the set of strict local minmax that are stable for $\dot{z} = -\Lambda_\tau g(z)$ almost surely. That is, for almost all initial conditions, $\tau$-GDA will converge to a strict local minmax.*

If we fix a $\tau$ satisfying the assumptions of the above theorem, then if we consider the set of stable critical points for $\dot{z} = -\Lambda_\tau g(z)$, we know that by Theorem 1 this set only contains differential Stackelberg (and hence, strict local minmax) points. It is precisely this set of strict local minmax to which $\tau$-GDA converges almost surely. Note that a stable differential Stackelberg equilibrium is ensured to exist for the choice of $\tau$ and $\gamma$. Moreover, by Theorem 1, as we increase $\tau$, no new spurious non-minmax points are introduced to the set of stable critical points; only additional strict local minmax points can be added to this set. Thus, this result provides a novel convergence guarantee to not just $\varepsilon$-stationary points but those that are game theoretically meaningful.

## 5.2 Convergence Rates to Approximate Stationary Points

While Theorem 2 implies a global convergence guarantee to a strict local minmax equilibrium, a global convergence rate to the same set does not easily follow. This is due to the fact that obtaining such a result requires the dynamics to escape strict saddle points not just asymptotically, but efficiently. In lieu of pursuing such a result, in this subsection, we derive global complexity bounds for $\tau$-GDA reaching an $\varepsilon$-critical point. To begin, we have the following convergence result for $\tau$-GDA, which follows rather directly from the potential function given in Lemma 1.

**Corollary 1.** *Consider a nonconvex-PŁ/SC zero-sum game $(f, -f)$ defined by $f \in C^2(\mathcal{Z}, \mathbb{R})$ that satisfying Assumption 1. Then, $\tau$-GDA from any initialization with $\tau \geq \Gamma^{-1}\kappa^2$, $\gamma < \min\{\frac{1}{2L}, \frac{1}{2\tau L}, \frac{1}{2L_4}\}$ for $\Gamma \in (0, 1/7]$, has at least one iterate that is an $\varepsilon$-critical point after $\widetilde{\mathcal{O}}(\varepsilon^{-2})$ iterations.*

We remark that we are unaware of an existing convergence result to a $\varepsilon$-critical point for $\tau$-GDA in nonconvex-PŁ zero-sum games. Moreover, it is worth noting that this convergence guarantee holds for choices of $\tau$ and $\gamma$ that satisfy the conditions of Theorem 2. Thus, together, we have shown that $\tau$-GDA finds an approximate stationary point efficiently, and then asymptotically is guaranteed to converge to a strict local minmax equilibrium. It is also known that $\tau$-GDA locally converges

---

[3]The strict saddle avoidance result of Mazumdar et al. [40, Theorem 4.1] holds more generally for simultaneous gradient descent with heterogeneous learning rates in $n$-player general-sum nonconvex games. For simplicity, we only state the result in the context of this work. We also remark that Daskalakis and Panageas [10, Theorem 2.2] show an analogous strict saddle avoidance result for $\tau$-GDA in nonconvex-nonconcave zero-sum games, but restricted to $\tau = 1$, hence the reason it is not referenced in Lemma 2.

exponentially fast around stable strict local minmax equilibrium in nonconvex-nonconcave zero-sum games [15] and thus in this class of games as well.

We now restrict our focus to nonconvex-SC zero-sum games and provide a complimentary convergence analysis. In the existing literature on convergence in the nonconvex-SC literature [35, 59, 62], the proposed potential function has a structure closely related to the potential function in Lemma 1; in particular, it takes the form $f(x, y) + r(x, y)$ where $r(x, y)$ is a tracking term of the form $\|y - y_*(x)\|^2$ or $f(x, y_*(x))$. Yet, perhaps surprisingly, despite the fact that the game cost depends on the sequences generated by two separate gradient updates—one from minimizing $f$ and one from maximizing it—we show below that there are choices of $\tau$ and $\gamma$ for $\tau$-GDA such that it is decreasing along trajectories of $\tau$-GDA. The benefit of knowing that the cost function is a potential is that it is a computable quantity that can be monitored to evaluate progress during training in machine learning problems formulated as zero-sum games in this class. For this set of results, we consider both the deterministic and stochastic settings. In the stochastic setting, for any $(x, y)$, a gradient query to an oracle returns $g_1(x, y; \theta_1)$ and $g_2(x, y; \theta_2)$ to the two players respectively, where each $\theta_i$ is a random variable drawn from distribution $\mathcal{D}_i$ with $i \in \mathcal{I}$ and the stochastic gradients satisfy the following assumptions.[4]

**Assumption 3.** *For any* $(x, y) \in \mathcal{Z}$ *and each* $i \in \mathcal{I}$, $\mathbb{E}_{\theta_i \sim \mathcal{D}_i}[g_i(x, y; \theta_i)] = \nabla_i f(x, y)$ *and* $\forall\, t \in \mathbb{R}$:

$$\mathbb{P}(\|g_i(x, y; \theta) - \nabla_i f(x, y)\| \geq t) \leq 2 \exp(-\tfrac{t^2}{2\sigma_i^2}).$$

Given this assumption and defining $\ell := L_1 + 3\kappa' L_2$ and $\kappa' := L_2/\mu$, we have the following result.

**Lemma 3.** *Consider a nonconvex-SC zero-sum game* $(f, -f)$ *defined by* $f \in C^2(\mathcal{Z}, \mathbb{R})$ *satisfying Assumption 1 and Assumption 3. If* $\gamma \leq 1/\ell$ *and* $\tau = \Gamma/(\mu\gamma)$ *for any constant* $\Gamma \geq 4$, *then with probability at least* $1 - \delta$ *for any* $\delta \in (0, 1)$, $f(x_k, y_k)$ *is a (stochastic) potential function for* $\tau$-GDA *in the stochastic and deterministic settings.*

This result then leads to the following convergence guarantee to $\varepsilon$-critical points, given that there is a finite minimum value $f^*$ of the game cost function.

**Corollary 2.** *Consider a nonconvex-SC zero-sum game* $(f, -f)$ *defined by* $f \in C^2(\mathcal{Z}, \mathbb{R})$ *satisfying Assumption 1 and Assumption 3. For any* $\delta \in (0, 1)$, *there exists* $\gamma$ *and* $\tau$ *satisying the conditions of Lemma 3 such that, with probability at least* $1 - \delta$, *starting from any initialization, at least half the iterates of (stochastic)* $\tau$-GDA *will be* $\varepsilon$-critical points after $\widetilde{\mathcal{O}}(\varepsilon^{-2})$ and $\widetilde{\mathcal{O}}(\varepsilon^{-6})$ *iterations in the deterministic and stochastic settings, respectively.*

## 6 Discussion

To the best of our knowledge, our results are the first to guarantee global convergence of gradient-based algorithms to game theoretically meaningful equilibria in nonconvex-PŁ/SC zero-sum games. We believe an interesting direction of future work in these classes of games would be to obtain an equivalent guarantee in the setting with constraints. Moreover, in the unconstrained setting, it would be desirable to show not only does a $\tau$-GDA variant provably converge in an almost sure asymptotic sense to strict local minmax equilibria, but that there is also finite-time guarantee (preferably matching the rates from nonconvex optimization) to an appropriate approximate notion. The difficulty in showing this stems from possibility that the game dynamics can get stuck in a region around a non-game theoretically meaningful strict saddle. This problem has been observed in the nonconvex single player setting [12] with gradient descent and trivially extends to $\tau$-GDA in the zero-sum game setting. A body of work in nonconvex optimization has developed algorithmic methods that address this issue [9, 14, 17, 20, 24] including by injecting artificial noise to perturb the updates away from strict saddle points. Yet, to our knowledge, such methods have not yet been extended to the zero-sum game setting. Notably, escaping strict saddles is more of nuanced issue in this setting since strict saddles of the dynamics are not always strict saddles of the function and vice versa.

## Acknowledgments and Disclosure of Funding

This work was supported by Office of Naval Research Young Investigator Program and NSF CAREER Award 1844729. Tanner Fiez was also supported in part by a NDSEG Fellowship.

---

[4]Note that we are overloading notation by using $g_1(x, y; \theta_1)$ and $g_2(x, y; \theta_2)$ to denote stochastic individual gradients for each player, whereas previously $g(x, y)$ has denoted the vector of each players gradients.

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
