# Appendix

Below we provide a table of contents as a guide to the appendix.

## Table of Contents

## A  Related Work

Appendix A contains an extensive discussion of related work on nonconvex-nonconcave zero-sum games, nonconvex-PŁ/SC zero-sum games, and escaping saddles points.

Appendix B contains technical preliminaries on game theory (Appendix B.1), dynamical systems (Appendix B.2), PŁ functions and nonconvex-PŁ zero-sum games (Appendix B.3), linear algebra (Appendix B.4), and concentration of measure (Appendix B.5). Appendix B.1 includes a formal local Stackelberg equilibrium definition that results in the differential Stackelberg (strict local minmax) equilibrium notion we consider. Appendix B.2 provides background on determining the stability of dynamical systems using the local linearization. Appendix B.3 contains useful properties of PŁ functions and nonconvex-PŁ zero-sum games that help in both the stability analysis in Appendix C and for proving the results from Section 5. Appendix B.4 contains a technical result that is needed for the stability analysis in Appendix C. Appendix B.5 includes concentration inequalities for norm-subGaussian random vectors needed for the stochastic convergence result in Appendix E.

Appendix C contains the proof of Theorem 1 from Section 4 regarding the local stability of critical points in nonconvex-PŁ zero-sum games.

Appendix D contains the proofs for the results from Section 5 on the convergence of $\tau$-GDA in nonconvex-PŁ zero-sum games, which also immediately generalize to nonconvex-SC zero-sum games. In particular, this Appendix D includes the proofs of Lemma 1 (potential function), Theorem 2 (global asymptotic equilibrium convergence guarantee) and Corollary 1 (convergence rate).

Appendix E contains the proofs for the results from Section 5 specific to nonconvex-SC zero-sum games. In particular, Appendix E contains the proofs of Lemma 3 (potential function) and Corollary 2 (convergence rates).

We now cover related work in further detail. We begin by discussing past work on nonconvex-nonconcave zero-sum games with a focus on papers that analyze the local stability of critical points using the continuous-time limiting system as we do in Section 4 for nonconvex-PŁ/SC zero-sum games. Then, we compare our convergence results in nonconvex-PŁ/SC zero-sum games presented in Section 5 to existing guarantees in the literature for various gradient-based learning algorithms. We remark that there are a number of papers analyzing gradient-based learning in strongly-convex-strongly-concave, strongly-convex-linear, strongly-convex-concave, convex-concave, and nonconvex-concave zero-sum games. Since each of the aforementioned problems are fundamentally different in terms of the structure compared to nonconvex-PŁ/SC zero-sum games, we do not discuss each body of work in detail and instead refer the reader to the detailed discussion of work on such problems provided in the paper of Lin et al. [34]. Finally, we review techniques and results for escaping saddle points in nonconvex optimization relevant to our analogous methods and guarantees in classes of nonconvex games.

**Nonconvex-Nonconcave Zero-Sum Games.**    In general nonconvex-nonconcave zero-sum games, global convergence guarantees to traditional game-theoretic equilibrium notions are effectively unobtainable as a result of computational hardness results [11] and the emergence of non-trivial limit cycles and periodic orbits [19, 30]. Consequently, the analysis of gradient-based learning algorithms in nonconvex-nonconcave zero-sum games has commonly focused on determining the local stability of the continuous-time limiting system around critical points. Typically, the goal is to determine the relationship between the set of stable critical points and the critical points that correspond to game-theoretic equilibrium.

The prototypical equilibrium notions in game theory are the (local) Nash and Stackelberg equilibrium concepts [6]. The aforementioned equilibrium notions have been characterized in terms of gradient-based sufficient conditions and

critical points satisfying the conditions have sometimes been termed differential Nash (strict local Nash) and differential Stackelberg (strict local Stackelberg) equilibria. The differential characterization of local Nash equilibria was reported by Ratliff et al. [49, 51], while the differential characterization of local Stackelberg equilibria was given concurrently by Fiez et al. [16] and Jin et al. [23] where the conditions presented by the former extend to general-sum games. Moreover, Fiez et al. [16] and Jin et al. [23] show that differential Nash equilibria are a subset of differential Stackelberg equilibria. A series of works analyze the properties of the equilibrium notions defined in terms of gradient-based sufficient conditions. In particular, Ratliff et al. [50] and Mazumdar and Ratliff [39] prove the genericity and structural stability of differential Nash equilibria in general-sum and zero-sum games, respectively. Analogously, Fiez et al. [16] prove the genericity and structural stability of differential Stackelberg equilibria in zero-sum games.

Several past works study the local stability of $\tau$-GDA without timescale separation (equal learning rates, $\tau = 1$) around critical points [10, 40]. In this direction, each of the aforementioned works prove there can exist stable critical points of $\tau$-GDA with $\tau = 1$ that are not differential Nash equilibria. Moreover, they each show that differential Nash equilibria are stable critical points of $\tau$-GDA with $\tau = 1$.

Building on this line of work, the local stability of $\tau$-GDA with timescale separation around critical points has been analyzed [15, 23]. We begin by commenting on the results of Jin et al. [23]. It is shown by the authors that differential Nash equilibria are stable critical points of $\tau$-GDA with any timescale separation parameter $\tau > 0$. Moreover, they prove that given any fixed timescale separation $\tau > 0$, there exists a game with a differential Stackelberg equilibrium that is not stable with respect to the $\tau$-GDA system. Finally, they show that as $\tau \to \infty$, the stable critical points of $\tau$-GDA coincide with the set of differential Stackelberg equilibria. Fiez and Ratliff [15] provide complimentary results by focusing on the stability properties of any given critical point as a function of the timescale separation $\tau$, rather than considering stability properties across the space of nonconvex-nonconcave zero-sum games. In particular, they show a stability and instability result. The stability result says that given a differential Stackelberg equilibrium $(x^*, y^*)$, there exists a finite $\tau^* \in (0, \infty)$ that can be constructed such that $(x^*, y^*)$ is stable for all $\tau \in (\tau^*, \infty)$ with respect to the $\tau$-GDA dynamics. The instability results says that given a critical point $(x^*, y^*)$ which is not a differential Stackelberg equilibrium, there exists a finite $\tau_0 \in (0, \infty)$ that can be constructed such that $(x^*, y^*)$ is not stable for all $\tau \in (\tau_0, \infty)$ with respect to the $\tau$-GDA dynamics. It is important to observe that the constructions of a $\tau^*$ or $\tau_0$ depend on the properties of a given critical point.

The results we provide regarding the local stability of $\tau$-GDA around critical points in nonconvex-PŁ/SC zero-sum games in Section 4 strengthen the stability characterizations of Fiez and Ratliff [15] by exploiting the structure of these classes of games. In comparison, we obtain the stronger results in nonconvex-PŁ zero-sum games that (i) $\tau_0 = 0$ for any critical point that is not a differential Stackelberg equilibria and (ii) a differential Stackelberg equilibria is never unstable after it becomes stable as a function of $\tau$. That is, we obtain insights that apply to the class of games rather than specific critical points, which are easily quantifiable instead of depending on properties of a given critical point.

We remark that another line of work studies the local stability of $\tau$-GDA without timescale separation ($\tau = 1$) in generative adversarial networks under certain assumptions [42, 43]. The results are generalized to the study of $\tau$-GDA with timescale separation [15] to show that equivalent conclusions hold for any choice of timescale separation $\tau > 0$.

Finally, there is a line of work developing gradient-based learning algorithms using second order information in nonconvex-nonconcave zero-sum games such that the only locally stable critical points correspond to game-theoretic equilibrium notions. In particular, Adolphs et al. [1] and Mazumdar et al. [41] develop algorithms under which a critical point is locally stable if and only if it is a differential Nash equilibrium. Moreover, Fiez et al. [16], Wang et al. [58], and Zhang et al. [64] construct algorithms under which a critical point is locally stable if and only if it is a differential Stackelberg equilibria.

**Nonconvex-PŁ and Nonconvex-SC Zero-Sum Games.** We now expand our discussion of existing work on nonconvex-PŁ and nonconvex-SC zero-sum games. As discussed in Section 2, in comparison to past works in this area, the key distinction of this work is that we give global convergence guarantees to game-theoretic equilibria, whereas existing work only considers convergence to notions of stationarity.

In the class of nonconvex-PŁ zero-sum games, the closest works to this paper are [45] and [62]. Each of the aforementioned works show that an $\varepsilon$-approximate stationary point of the game function $f(\cdot, \cdot)$ can be found in $\widetilde{\mathcal{O}}(\varepsilon^{-2})$ gradient calls by variants of alternating gradient descent-ascent with timescale separation ($\tau$-AGDA) in the deterministic setting. In comparison, we show in Theorem 2 that simultaneous gradient descent-ascent with timescale separation ($\tau$-GDA) globally asymptotically converges to a differential Stackelberg equilibrium (local minmax equilibrium) almost surely. Moreover, we also provide a global convergence guarantee to an $\varepsilon$-critical point in $\widetilde{\mathcal{O}}(\varepsilon^{-2})$ gradient calls in the deterministic setting. We remark that the primary focus of [62] is on two-sided PŁ games, which is a class with a unique equilibrium and the convergence guarantees given for this class are to the equilibrium.

In the class of nonconvex-SC zero-sum games, there is an extensive amount of recent work. In terms of existing results on variants of gradient descent-ascent, Lin et al. [33] shows for both simultaneous and alternating gradient descent-ascent ($\tau$-GDA and $\tau$-AGDA) that with $\widetilde{\mathcal{O}}(\varepsilon^{-2})$ and $\widetilde{\mathcal{O}}(\varepsilon^{-5})$ gradient calls an $\varepsilon$–approximate stationary point

of the function $f(\cdot, \cdot)$ or the function $\phi(\cdot) = \max_{y \in \mathcal{Y}} f(\cdot, y)$ can be obtained in the deterministic and stochastic settings, respectively. For this class of games, we show in Theorem 2 that $\tau$-GDA globally asymptotically converges to a differential Stackelberg equilibrium (local minmax equilibrium) almost surely. Obtaining this result requires both fully characterizing the local stability around critical points (Theorem 1) and then combining asymptotic saddle escape results with a potential function. Moreover, we also show that there exists $\tau$ and $\gamma$ such that the game cost function is a potential function and using this derive convergence rates to $\varepsilon$-critical points of $\widetilde{\mathcal{O}}(\varepsilon^{-2})$ and $\widetilde{\mathcal{O}}(\varepsilon^{-6})$ in deterministic and stochastic settings, respectively.

Before moving on, we remark that there is a number of works that develop algorithms that improve the complexity of finding stationary points in terms of the dependence on the condition number and polylogarithmic dependencies. This focus is separate from our work, but we refer the reader to [34, 36, 48] and the references therein. We believe an interesting direction of future work is strengthening the results along this direction to the stronger notions of solving the problem considered in this work. Furthermore, there is also recent work developing lower bounds for this class of games [32, 65]. Finally, there are several works in this class of games with zeroth order feedback [35, 59, 60].

**Escaping Saddle Points.**    In this paper we present asymptotic convergence results to differential Stackelberg equilibrium in nonconvex-PŁ and nonconvex-SC zero-sum games. To obtain such results it is necessary to escape saddle points asymptotically. We build on methods from nonconvex-nonconcave games and nonconvex optimization to prove that $\tau$-GDA has this guarantee. In what follows, we provide background on these methods.

A common assumption in the body of work on escaping saddle points in nonconvex optimization is what is known as the strict saddle property (see, e.g., [17, 20, 28]). Informally, if a function class satisfies the strict saddle property, every saddle point of the gradient descent dynamics has a strictly negative eigenvalue in the Jacobian evaluated at the critical point. This assumption ensures that there is a direction of escape from every saddle point. We make an analogous assumption adapted to the classes of nonconvex zero-sum games we analyze (Assumption 2) following the definition from the work of Mazumdar et al. [40]. For nonconvex-PŁ/SC zero-sum games, if the game satisfies the strict saddle property, every saddle point of the $\tau$-GDA dynamics has an eigenvalue with strictly negative real part and no eigenvalue has a real part equal to zero in the Jacobian $J_\tau(x^*, y^*)$ evaluated at the critical point for the given choice of timescale separation $\tau$. We remark that there is some distinction between the definition of a strict saddle between nonconvex optimization and nonconvex games. This stems from the fact that in nonconvex optimization, saddles of the function and saddles of the dynamics are equivalent, whereas this does not carry over to the zero-sum game setting. In this paper, the terminology of a saddle point is with respect to the dynamics.

There exists both asymptotic and finite-time saddle avoidance results in nonconvex optimization. The asymptotic saddle avoidance results in nonconvex optimization state that gradient descent dynamics almost surely avoid strict saddle points [28, 29]. This result has been extended to show that gradient descent dynamics almost surely avoid saddle points even for functions with non-isolated critical points [46] in nonconvex optimization. The drawback of this style of result is that it fails to preclude that gradient descent could spend an arbitrarily long time stuck in the neighborhood of a saddle point. In fact, it has been shown that gradient descent can take exponential time to escape from saddle points [12].

A series of works present asymptotic results on escaping saddle points in continuous action games analogous to that from nonconvex optimization. In particular, Mazumdar et al. [40] prove that in $N$-player general-sum games, if each player follows the gradient descent learning rule then the dynamics avoid strict saddles of the dynamics almost surely. This result allows for players to employ distinct learning rates. Translated to nonconvex-nonconcave zero-sum games, the result ensures that $\tau$-GDA dynamics avoid strict saddles of the dynamics almost surely for any timescale separation $\tau > 0$ as is presented in Lemma 2. The aforementioned result is key to the asymptotic convergence guarantee to differential Stackelberg equilibria we provide in Theorem 2 for $\tau$-GDA for nonconvex-PŁ/SC zero-sum games. In a related line of work, Daskalakis and Panageas [10] show that in nonconvex-nonconcave zero-sum games the $\tau$-GDA dynamics with $\tau = 1$ (without timescale separation) avoid strict saddle points including when critical points are not isolated. We remark that the result demonstrating that gradient descent can take exponential time to escape from saddle points [12] immediately carries over to gradient descent-ascent in zero-sum games since the game could be completely decoupled an correspond to separate nonconvex-optimization problems.

The study of finite-time saddle avoidance results in nonconvex optimization generally focuses on designing variants of gradient descent that not only avoid saddle points of the dynamics almost surely, but escape from them efficiently. The methods of greatest relevance to this paper are the existing works analyzing the rates at which 'simple' variants of gradient descent escape saddle points in nonconvex optimization. This line of research dates back to the work of Ge et al. [17], who proved that stochastic gradient descent with injected noise perturbations finds an approximate local minimum in a number of gradient class that depends polynomially on the dimension. Since the initial work on escaping saddle points efficiently, there have been many follow-up works. The most relevant to our results and techniques are the works of Jin et al. [20] and Jin et al. [24]. Indeed, Jin et al. [20] prove that a perturbed variant of gradient descent with deterministic gradients finds approximate local minima under the strict saddle assumption with only a logarithmic dimension dependence, meaning the result is almost dimension free and near equivalent to the complexity needed to

find a critical point using gradient descent. Moreover, Jin et al. [24] generalize such proof techniques to show that variants of gradient and stochastic gradient descent with injected noise achieve an analogous guarantee with a unified analysis. For future work, it would be interesting to see if these methods could be extended to a perturbed variant of $\tau$-GDA toward seeking a finite-time convergence guarantee to an approximate local minmax equilibrium. There is an extensive literature on methods for escaping saddle points efficiently in nonconvex optimization beyond the works that have been mentioned thus far. This includes analysis of dynamics using normalized gradients [31], stochastic gradient descent without artificial noise injections [9], negative curvature search methods [2–4, 61], variance reduced methods [13, 66], cubic regularization [56], and adaptive acceleration methods [7, 21, 54, 57]. Finally, there is also a recent, sharper analysis of stochastic gradient descent [14].

# B  Preliminaries

We now review preliminaries on game-theory concepts, dynamical systems theory, PŁ functions and nonconvex-PŁ zero-sum games, linear algebra, and concentration inequalities. The sections on PŁ functions and nonconvex-PŁ zero-sum games, linear algebra, and concentration inequalities include a number of technical lemmas that are needed throughout the rest of the appendix.

## B.1  Game Theory

The focus of this work is on developing convergence guarantees for gradient descent-ascent with timescale separation ($\tau$-GDA) to game-theoretically meaningful equilibrium. In the class of games under consideration, the natural solution concept from game theory is a minmax or equivalently Stackelberg equilibrium. Given the nonconvex nature of the minimizing players objective, we consider a local refinement of this historically standard equilibrium concept. In the nonconvex-nonconcave zero-sum game literature, recent work [16, 23] develops sufficient conditions for local Stackelberg equilibrium and points satisfying the conditions correspond to strict local Stackelberg equilibrium. We refer to strict local Stackelberg equilibrium as differential Stackelberg equilibrium following the terminology of [16]. This is the the basis for the equilibrium concept we consider that is given in Definition 4. The fact we focus our attention on strict local equilibrium is standard [15, 16, 23, 40, 58]).

For completeness, we now present a formal local Stackelberg equilibrium definition characterized in terms of the costs. This definition has appeared in past work [15, 16] and it is a simple refinement to a local concept from the standard Stackelberg equilibrium definition in continuous action space games [6]. We emphasis that the equilibrium we consider in Definition 4 is characterized by sufficient conditions for the following local Stackelberg equilibrium definition.

**Definition 7** (Local Minmax/Stackelberg Equilibrium). *Consider $U_x \subset \mathcal{X}$ and $U_y \subset \mathcal{Y}$ where, without loss of generality, player 1 (controlling $x \in \mathcal{X}$) is minimizing player and player 2 (controlling $y \in \mathcal{Y}$) is the maximizing player. The strategy $x^* \in U_x$ is a local Stackelberg solution for the minimizing player if, $\forall x \in U_x$,*

$$\sup_{y \in r_{U_y}(x^*)} f(x^*, y) \leq \sup_{y \in r_{U_y}(x)} f(x, y),$$

*where $r_{U_y}(x) = \{y' \in U_y | f(x, y') \geq f(x, y), \forall y \in U_y\}$ is the reaction curve. Moreover, for any $y^* \in r_{U_y}(x^*)$, the joint strategy profile $(x^*, y^*) \in U_x \times U_y$ is a local Stackelberg equilibrium on $U_x \times U_y$.*

## B.2  Dynamical Systems Theory

In this section, we provide relevant background as it pertains to the local stability analysis we presented in Section 4.

We begin by formally describing what is meant by local stability of a continuous-time dynamical system as used in this paper. In particular, we provide a formal definition for the stability definition that was presented in Definition 5. The following well-known result provides equivalent characterizations of local stability for a critical point (stationary point) of a continuous-time dynamical system of the form $\dot{z} = -g(z)$ in terms of the Jacobian matrix $J(z) = Dg(z)$.

**Theorem 3** ([26, Theorem 4.6, Corollary 4.3]). *Consider a critical point $z^*$ of $g(z)$. The following are equivalent: (a) $z^*$ is a locally exponentially stable equilibrium of $\dot{z} = -g(z)$; (b) $\mathrm{spec}(-J(z^*)) \subset \mathbb{C}_-^\circ$; (c) there exists a symmetric positive-definite matrix $P = P^\top \succ 0$ such that $P J(z^*) + J(z^*)^\top P \succ 0$.*

Before moving on, we provide a brief discussion of the implications of determining stability using the local linearization (Jacobian of the dynamical system) around critical points. The Hartman-Grobman theorem [52, Theorem 7.3]; [55, Theorem 9.9] asserts that it is possible to continuously deform all trajectories of a nonlinear system onto trajectories of the linearization at a fixed point of the nonlinear system. Informally, the theorem states that the qualitative properties of the nonlinear system $\dot{z} = -g(z)$ in the vicinity (which is determined by the neighborhood $U$) of an isolated equilibrium $z^*$ are determined by its linearization if the linearization has no eigenvalues on the imaginary axes in the complex plane. We also remark that Hartman-Grobman can also be applied to discrete time maps (cf. Sastry [52, Thm. 2.18]) with the same qualitative outcome.

In the context of this work, this means that by determining stability using the local linearization around critical points, the behavior of the nonlinear system in a neighborhood of the critical point can be inferred. In particular, given that a critical point is determined to be stable by the local linearization, then there is a neighborhood on which the dynamics converge to the critical point. This observation also applies to the discrete-time system with proper learning rates.

### B.3 Polyak-Łojasiewicz Functions and Nonconvex-Polyak-Łojasiewicz Zero-Sum Games

In this section, we state properties of PŁ functions in the context of nonconvex-PŁ zero-sum games. Specifically, we state additional properties of nonconvex-PŁ zero-sum games that follow from properties of PŁ functions (see Lemma 4 and Lemma 5). Moreover, we present additional smoothness properties (see Lemma 6 and Lemma 7) that follow from properties of nonconvex-PŁ zero-sum games and Assumption 1. Finally, we characterize the curvature around critical points of PŁ functions (see Lemma 8). These properties will be used in both the proofs for the local stability and the global convergence in nonconvex-PŁ zero-sum games.

We begin by stating a known property [25] that $\mu$-PŁ functions satisfy a quadratic growth condition [5] also with parameter $\mu$. For clarity of presentation, we state this condition in the context of the nonconvex-PŁ zero-sum games we study. Recall from Definition 1 that given a zero-sum game $(f, -f)$ defined by $f \in C^2(\mathcal{Z}, \mathbb{R})$, the game is called nonconvex-PŁ if $f(x, \cdot)$ is $\mu$-PŁ with respect to the argument $y \in \mathcal{Y}$. That is, for $\mu > 0$ and for all $z = (x, y) \in \mathcal{Z}$,

$$\|\nabla_2 f(x, y)\|^2 \geq 2\mu(\max_{y' \in \mathcal{Y}} f(x, y') - f(x, y)), \tag{7}$$

where for any fixed $x \in \mathcal{X}$, $\max_{y' \in \mathcal{Y}} f(x, y')$ has a nonempty solution set and a finite optimal value.

**Lemma 4** ([25, Theorem 2, Appendix A]). *Consider a nonconvex, $\mu$-PŁ zero-sum game defined by $f \in C^2(\mathcal{Z}, \mathbb{R})$ satisfying Assumption 1. For all $x \in \mathcal{X}$, the function $f(x, \cdot)$ satisfies the following quadratic growth condition:*

$$\max_{y' \in \mathcal{Y}} f(x, y') - f(x, y) \geq \frac{\mu}{2} \|y_p - y\|^2, \ \forall \, y \in \mathcal{Y} \tag{8}$$

*where $y_p$ is the projection onto the set $\operatorname{argmax}_{y \in \mathcal{Y}} f(x, y)$.*

It is also known that $\mu$-PŁ functions satisfy an error bound condition [37] also with parameter $\mu$. Again, for clarity of presentation, we state this condition in the context of the nonconvex-PŁ zero-sum games we study.

**Lemma 5** ([25, Theorem 2, Appendix A]). *Consider a nonconvex, $\mu$-PŁ zero-sum game defined by $f \in C^2(\mathcal{Z}, \mathbb{R})$ satisfying Assumption 1. For all $x \in \mathcal{X}$, the function $f(x, \cdot)$ satisfies the following error bound condition:*

$$\|\nabla_2 f(x, y)\| \geq \mu \|y_p - y\|, \ \forall \, y \in \mathcal{Y} \tag{9}$$

*where $y_p$ is the projection onto the set $\operatorname{argmax}_{y \in \mathcal{Y}} f(x, y)$.*

We now state additional smoothness properties that follow from properties of nonconvex-PŁ zero-sum games and Assumption 1. These properties are known for this class of games. For nonconvex-PŁ zero-sum games, given any fixed $x \in \mathcal{X}$, the set of maximizers of $f(x, \cdot)$ may not be a singleton. For any fixed $x \in \mathcal{X}$, we denote by $y_*(x)$ an element of $\operatorname{argmax}_{y \in \mathcal{Y}} f(x, y)$. When necessary, we point out to which specific element $y_*(x)$ corresponds. The following result shows that the mapping $y_*(x)$ satisfies a stability property.

**Lemma 6** (Stability of Best-Response Map in PŁ functions [45, Lemma A.3]). *Consider a nonconvex, $\mu$-PŁ zero-sum game defined by $f \in C^2(\mathcal{Z}, \mathbb{R})$ satisfying Assumption 1 and let $L_3 := L_2/\mu$. For all $x, x' \in \mathcal{X}$ and $y_*(x) \in \operatorname{argmax}_{y \in \mathcal{Y}} f(x, y)$, there exists a $y_*(x') \in \operatorname{argmax}_{y \in \mathcal{Y}} f(x, y)$ such that*

$$\|y_*(x) - y_*(x')\| \leq L_3 \|x - x'\|.$$

Danskin's theorem in optimization provides conditions under which the total derivative $\nabla f(x, y_*(x))$ where

$$y_*(x) \in \operatorname*{argmax}_{y \in \mathcal{Y}} f(x, y)$$

is equivalent to $\nabla_1 f(x, y_*(x))$. That is, it gives conditions when the gradient of the function $f(x, y_*(x))$ is equal to the gradient of $f(x, y)|_{y=y_*(x)}$ evaluated directly at the optimum. Typically this requires the maximizer to be unique. However, it has been shown that for nonconvex-PŁ zero-sum games, this property carries over even without a unique solution. This is stated in the following result along with a smoothness property of the function $\max_{y \in \mathcal{Y}} f(x, y)$.

**Lemma 7** (Danskin-Type Property for PŁ functions [45, Lemma A.5]). *Consider a nonconvex, $\mu$-PŁ zero-sum game defined by $f \in C^2(\mathcal{Z}, \mathbb{R})$ satisfying Assumption 1. Then,*

$$\nabla f(x, y_*(x)) = \nabla_1 f(x, y_*(x)) \quad \text{where} \quad y_*(x) \in \operatorname*{argmax}_{y \in \mathcal{Y}} f(x, y).$$

*Moreover, $\nabla f(x, y_*(x))$ is $L_4 = (L_1 + \frac{L_1 L_2}{\mu})$–Lipschitz. That is, for all $x, x' \in \mathcal{X}$,*

$$\|\nabla f(x, y_*(x)) - \nabla f(x', y_*(x'))\| \leq L_4 \|x - x'\|.$$

We remark that $L_3$ and $L_4$ from Lemma 6 and Lemma 7 slightly deviate from that values in the work of Nouiehed et al. [45]. This is a result of the fact that a correction was made in the work of Karimi et al. [25] after the publication of the work of Nouiehed et al. [45], which showed that $\mu$-PŁ functions satisfy the quadratic growth condition with parameter $\mu$ whereas previously it was stated that $\mu$-PŁ functions satisfy the quadratic growth condition with parameter $4\mu$. Consequently, $L_3$ and $L_4$ are stated in Lemma A.3 and Lemma A.5 in the work of Nouiehed et al. [45] (in our notation) as $L_2/(2\mu)$ and $L_1 + (L_1L_2)/(2\mu)$, respectively. Incorporating the fix to the constant from the most recent version of Karimi et al. [25] into Lemma A.3 and Lemma A.5 in the work of Nouiehed et al. [45] gives the values of $L_3$ and $L_4$ stated in Lemma 6 and Lemma 7, respectively.

We now show that the quadratic growth property of PŁ functions implies that the eigenvalues of $\nabla_2^2 f(x^*, y^*)$ are upper-bounded by $-\mu$ at any critical point $(x^*, y^*)$ of a $\mu$-PŁ zero-sum game. This means that at any critical point of the game, player 2 must be at a maximum. It is in fact known that all critical points of PŁ functions are global minimum [25], which in the context of this work implies that all critical points of nonconvex-PŁ zero-sum games, player 2 must be not only at a local maximum, but a global maximum. This can be observed by the definitions of a critical point and nonconvex-PŁ zero-sum games we have at any critical point $(x^*, y^*)$:

$$0 = \|\nabla_2 f(x^*, y^*)\|^2 \geq 2\mu(\max_{y' \in \mathcal{Y}} f(x^*, y') - f(x^*, y^*)) \geq 0.$$

The following property will be used in the proof of Theorem 1 in Appendix C. Note that given the inequality above it is primary needed to show that $\det(\nabla_2^2 f(x^*, y^*)) \neq 0$ at critical points $(x^*, y^*)$.

**Lemma 8.** *Consider a nonconvex, $\mu$-PŁ zero-sum game defined by $f \in C^2(\mathcal{Z}, \mathbb{R})$. At any critical point $(x^*, y^*)$ of the game, that is where $\nabla_1 f(x^*, y^*) = 0$ and $\nabla_2 f(x^*, y^*) = 0$, the individual Hessian of the maximizing player given by $\nabla_2^2 f(x^*, y^*)$ is negative definite and eigenvalues bounded above by $-\mu$.*

*Proof.* Let us consider any critical point $(x^*, y^*)$ of the game so that $\nabla_1 f(x^*, y^*) = 0$ and $\nabla_2 f(x^*, y^*) = 0$. Taking a Taylor expansion of $-f(x^*, \cdot)$ about the point $y^*$, we and get that

$$-f(x^*, y) = -f(x^*, y^*) - \nabla_2 f(x^*, y^*)^\top (y - y^*) - \frac{1}{2} \int_{y^*}^z (y - z)^\top \nabla_2^2 f(x^*, z)(y - z)dz.$$

This is equivalent to

$$-f(x^*, y) = -f(x^*, y^*) - \nabla_2 f(x^*, y^*)^\top (y - y^*)$$
$$- \frac{1}{2} \int_{\tau=0}^1 (y^* + \tau(y - y^*) - y)^\top \nabla_2^2 f(x^*, y^* + \tau(y - y^*))(y^* + \tau(y - y^*) - y)d\tau$$

Let $B_{y^*}(\|y - y^*\|)$ be a ball of radius $\|y - y^*\|$ centered at $y^*$. Then,

$$-f(x^*, y) = -f(x^*, y^*) - \nabla_2 f(x^*, y^*)^\top (y - y^*)$$
$$- \frac{1}{2} \int_{\tau=0}^1 (y^* + \tau(y - y^*) - y)^\top \nabla_2^2 f(x^*, y^* + \tau(y - y^*))(y^* + \tau(y - y^*) - y)d\tau$$
$$\leq -f(x^*, y^*) - \nabla_2 f(x^*, y^*)^\top (y - y^*) - \frac{1}{2} \max_{z \in B_{y^*}(\|y-y^*\|)} (z - y^*)^\top \nabla_2^2 f(x^*, z)(z - y^*)$$

so that

$$f(x^*, y) \geq f(x^*, y^*) + \frac{1}{2} \max_{z \in B_{y^*}(\|y-y^*\|)} (z - y^*)^\top \nabla_2^2 f(x^*, z)(z - y^*)$$
$$\geq f(x^*, y^*) + \frac{1}{2}(z - y^*)^\top \nabla_2^2 f(x^*, z)(z - y^*), \ \forall z \in B_{y^*}(\|y - y^*\|).$$

Hence, by the quadratic growth property of PŁ functions from Lemma 4,

$$\frac{\mu}{2}\|y - y^*\| \leq f(x^*, y^*) - f(x^*, y) \leq -\frac{1}{2}(z - y^*)^\top \nabla_2^2 f(x^*, z)(z - y^*), \ \forall z \in B_{y^*}(\|y - y^*\|)$$

which gives the desired result.

$\square$

### B.4 Linear Algebra

In this section, we state a property regarding matrix inertia that is important for the proof of Theorem 1 in Appendix C. The following result from Lancaster and Tismenetsky [27, Theorem 2, Chapter 13.1] is needed for the proof of Theorem 1 given in Appendix C. We include it here for ease of reference. For a given matrix $A \in \mathbb{R}^{n \times n}$, $\upsilon_+(A)$, $\upsilon_-(A)$, and $\zeta(A)$ are the number of eigenvalues of the argument that have positive, negative and zero real parts, respectively.

**Lemma 9** ([27, Theorem 2, Chapter 13.1]). *Consider a matrix $A \in \mathbb{R}^{n \times n}$.*

(a) *If $P$ is a symmetric matrix such that $AP + PA^\top = Q$ where $Q = Q^\top \succ 0$, then $P$ is nonsingular and $P$ and $A$ have the same inertia, meaning that*

$$\upsilon_+(A) = \upsilon_+(P), \; \upsilon_-(A) = \upsilon_-(P), \; \zeta(A) = \zeta(P). \tag{10}$$

(b) *On the other hand, if $\zeta(A) = 0$, then there exists a matrix $P = P^\top$ and a matrix $Q = Q^\top \succ 0$ such that $AP + PA^\top = Q$, and $P$ and $A$ have the same inertia (i.e., (10) holds).*

### B.5 Concentration Inequalities

In this section, we presentation concentration inequalities for norm-subGaussian random vectors. Each of the following technical lemmas are from the works of Jin et al. [22, 24] and we reproduce them here for clarity of presentation and easy reference. These concentration inequalities will be used in Appendix E to obtain Lemma 3 and Corollary 2 from Section 5.2.

The following defines a norm-subGaussian random vector.

**Definition 8.** *A random vector $x \in \mathbb{R}^d$ is norm-subGaussian if there exists $\sigma$ so that:*

$$\mathbb{P}(\|x - \mathbb{E}[x]\| \geq t) \leq 2e^{-\frac{-t^2}{2\sigma^2}} \quad \forall\, t \in \mathbb{R}.$$

The next result shows that a bounded random vector and a subGaussian random vector are special cases of a norm-subGaussian random vector.

**Lemma 10.** *There exists an absolute constant $c$ so that the following random vectors are $c\sigma$-norm-subGaussian:*

1. *A bounded random vector $x \in \mathbb{R}^d$ such that $\|x\| \leq \sigma$.*
2. *A random vector $x \in \mathbb{R}^d$, where $x = \psi e_1$ and the random variable $\psi \in \mathbb{R}$ is $\sigma$-subGaussian.*
3. *A random vector $x \in \mathbb{R}^d$ that is $(\sigma/\sqrt{d})$-subGaussian.*

We now define the properties of norm-subGaussian martingale difference sequences.

**Condition 1.** *Consider random vectors $x_1, \ldots, x_n \in \mathbb{R}^d$ and the corresponding filtrations $\mathcal{F}_i = \sigma(x_1, \ldots, x_i)$ for $i \in [n]$ such that $x_i | \mathcal{F}_{i-1}$ is zero-mean $\sigma_i$-norm-subGaussian with $\sigma_i \in \mathcal{F}_{i-1}$. That is:*

$$\mathbb{E}[X_i | \mathcal{F}_{i-1}] = 0, \quad \mathbb{P}(\|x_i\| \geq t | \mathcal{F}_{i-1}) \leq 2e^{-\frac{t^2}{2\sigma_i^2}}, \quad \forall t \in \mathbb{R}, \forall i \in [n].$$

The next results give concentration inequalities for the sum of norm squares of norm-subGaussian random vectors and the sum of inner products of norm-subGaussian random vectors with another set of random vectors.

**Lemma 11.** *Given random vectors $x_1, \ldots, x_n \in \mathbb{R}^d$ that satisfy condition 1 with fixed $\sigma_1 = \cdots = \sigma_n = \sigma$, then for any $\iota > 0$ there exists an absolute constant $c$ such that with probability at least $1 - e^{-\iota}$:*

$$\sum_{i=1}^{n} \|x_i\|^2 \leq c\sigma^2(n + \iota).$$

**Lemma 12.** *Given random vectors $x_1, \ldots, x_n \in \mathbb{R}^d$ that satisfy condition 1 and random vectors $\{u_i\}$ that satisfy $u_i \in \mathcal{F}_{i-1}$ for all $i \in [n]$, then for any $\iota > 0$ and $\lambda > 0$ there exists an absolute constant $c$ such that with probability at least $1 - e^{-\iota}$:*

$$\sum_{i=1}^{n} \langle u_i, x_i \rangle \leq c\lambda \sum_{i=1}^{n} \|u_i\|^2 \sigma_i^2 + \frac{1}{\lambda}\iota.$$

## C  Stability Analysis: Proof of Theorem 1

This appendix is devoted to providing the proof of Theorem 1. For ease of reference, we restate the result now before providing the proof.

**Theorem 1.** *Consider a nonconvex-PŁ/SC zero-sum game $(f, -f)$ where $f \in C^2(\mathcal{Z}, \mathbb{R})$. Then, the following hold: 1) Any critical point $z^* \in \mathcal{Z}$ that is not a differential Stackelberg equilibrium is unstable with respect to $\tau$-GDA for all $\tau \in (0, \infty)$; 2) If $z^*$ is a differential Stackelberg equilibrium, then $\mathrm{spec}(-J_\tau(z^*)) \subset \mathbb{C}_-^\circ$ for all $\tau \in [\tau_*, \infty)$ where $\tau_*$ is the minimum $\tau \in (0, \infty)$ such that $\mathrm{spec}(-J_\tau(z^*)) \subset \mathbb{C}_-^\circ$ and a finite $\tau_*$ is guaranteed to exist.*

*Proof of Theorem 1.* We begin by proving the first claim of the theorem statement.

**Proof of 1.** Let us first consider the case that a given a critical point $z^*$ could be such that $\mathsf{S}_1(J_1(z^*))$ or $-\nabla_2^2 f(z^*)$ are singular. By Lemma 8 we know that for any critical point $z^*$, $-\nabla_2^2 f(z^*) \succ 0$ so that $-\nabla_2^2 f(z^*)$ is non-singular. Observe that at any critical point $z^*$, since $-\tau \nabla_2^2 f(z^*)$ is positive definite for all $\tau \in (0, \infty)$, the following identity holds for any $\tau \in (0, \infty)$:

$$\det(J_\tau(z)) = \det(\mathsf{S}_1(J(z^*))) \det(-\tau \nabla_2^2 f(z^*)).$$

From the fact that $\det(-\tau \nabla_2^2 f(z^*))$, it then easily follows that $\det(J_\tau(z^*)) = 0$ if and only if $\det(\mathsf{S}_1(J(z^*))) = 0$. Note that $\det(J_\tau(z^*)) = 0$ if and only if $0 \in \mathrm{spec}(J_\tau(z^*))$ since eigenvalues of a real square matrix are either purely real or come in complex conjugate pairs. Hence, given any critical point such that $\det(\mathsf{S}_1(J(z^*))) = 0$, then $0 \in \mathrm{spec}(J_\tau(z^*))$ and $\mathrm{spec}(-J_\tau(z^*)) \not\subset \mathbb{C}_-^\circ$ for all $\tau \in (0, \infty)$. Hence, any critical point such that $\mathsf{S}_1(J_1(z^*))$ is singular is a saddle point and thus not stable for all $\tau \in (0, \infty)$ and such a point is not a differential Stackelberg equilibrium.

Now, suppose that $z^*$ is a critical point such that $\mathsf{S}_1(J_1(z^*))$ and $-\nabla_2^2 f(z^*)$ are non-singular. Let $\mathrm{spec}(-J_{\tau_0}(z^*)) \subset \mathbb{C}_-^\circ$ for some $\tau_0 \in (0, \infty)$. We know that $-\nabla_2^2 f(z^*) \succ 0$ by Lemma 8. We argue by contradiction that $\mathsf{S}_1(J_1(z^*)) \succ 0$. Towards this end, suppose not. That is, assume for the sake of contradiction that $\mathsf{S}_1(J_1(z^*))$ has at least one negative eigenvalue, or equivalently that $-\mathsf{S}_1(J_1(z^*))$ has at least one positive eigenvalue.

Since $\det(\mathsf{S}_1(J_1(z^*))) \neq 0$ and $\det(-\nabla_2^2 f(z^*)) \neq 0$, by Lemma 9.b, there exists non-singular Hermitian matrices $P_1, P_2$ and positive definite Hermitian matrices $Q_1, Q_2$ such that $-\mathsf{S}_1(J_1(z^*))P_1 - P_1 \mathsf{S}_1(J_1(z^*)) = Q_1$ and $\nabla_2^2 f(z^*)P_2 + P_2 \nabla_2^2 f(z^*) = Q_2$.

Furthermore, $-\mathsf{S}_1(J_1(z^*))$ and $P_1$ have the same inertia, meaning

$$\upsilon_+(-\mathsf{S}_1(J_1(z^*))) = \upsilon_+(P_1), \ \upsilon_-(-\mathsf{S}_1(J_1(z^*))) = \upsilon_-(P_1), \ \zeta(-\mathsf{S}_1(J_1(z^*))) = \zeta(P_1)$$

where for a given matrix $A$, $\upsilon_+(A), \upsilon_-(A)$, and $\zeta(A)$ are the number of eigenvalues of the argument that have positive, negative and zero real parts, respectively. Similarly, $\nabla_2^2 f(z^*)$ and $P_2$ have the same inertia:

$$\upsilon_+(\nabla_2^2 f(z^*)) = \upsilon_+(P_2), \ \upsilon_-(\nabla_2^2 f(z^*)) = \upsilon_-(P_2), \ \zeta(\nabla_2^2 f(z^*)) = \zeta(P_2).$$

Since $-\mathsf{S}(J_1(z^*))$ has at least one strictly positive eigenvalue by assumption for the sake of contradiction, $\upsilon_+(P_1) = \upsilon_+(-\mathsf{S}(J_1(z^*))) \geq 1$.

Define

$$P = \begin{bmatrix} I & L_0^\top \\ 0 & I \end{bmatrix} \begin{bmatrix} P_1 & 0 \\ 0 & P_2 \end{bmatrix} \begin{bmatrix} I & 0 \\ L_0 & I \end{bmatrix} \tag{11}$$

where $L_0 = (\nabla_2^2 f(z^*))^{-1} \nabla_{12}^\top f(z^*)$. Since $P$ is congruent to $\mathrm{blockdiag}(P_1, P_2)$, by Sylvester's law of inertia [18, Thm. 4.5.8], $P$ and $\mathrm{blockdiag}(P_1, P_2)$ have the same inertia, meaning that

$$\upsilon_+(P) = \upsilon_+(\mathrm{blockdiag}(P_1, P_2)), \upsilon_-(P) = \upsilon_-(\mathrm{blockdiag}(P_1, P_2)), \zeta(P) = \zeta(\mathrm{blockdiag}(P_1, P_2)).$$

Consider the matrix equation $-PJ_{\tau_0}(z^*) - J_{\tau_0}^\top(z^*)P = Q_{\tau_0}$ for $-J_{\tau_0}(z^*)$ where

$$Q_{\tau_0} = \begin{bmatrix} I & L_0^\top \\ 0 & I \end{bmatrix} B_{\tau_0} \begin{bmatrix} I & 0 \\ L_0 & I \end{bmatrix} \tag{12}$$

with

$$B_{\tau_0} = \begin{bmatrix} Q_1 & P_1 \nabla_{12} f(z^*) - \mathsf{S}(J_1(z^*))L_0^\top P_2 \\ (P_1 \nabla_{12} f(z^*) - \mathsf{S}(J_1(z^*))L_0^\top P_2)^\top & P_2 L_0 \nabla_{12} f(z^*) + (P_2 L_0 \nabla_{12} f(z^*))^\top + \tau_0 Q_2 \end{bmatrix}$$

which can be verified by straightforward calculations. The matrix $B_{\tau_0}$ is a symmetric matrix, and it is positive definite. Indeed, first observe that $Q_1 \succ 0$ and $Q_2 \succ 0$. Then showing $B_{\tau_0} \succ 0$ reduces to showing

$$P_2 L_0 \nabla_{12} f(z^*) + (P_2 L_0 \nabla_{12} f(z^*))^\top \succeq 0.$$

To see this, note that it is sufficient to choose $P_2 = \nabla_2^2 f(z^*) \prec 0$ so that

$$Q_2 = P_2 \nabla_2^2 f(z^*) + \nabla_2^2 f(z^*) P_2 = 2\nabla_2^2 f(z^*)^2 \succeq 2\mu^2 I.$$

Observe that we have used the fact that the eigenvalues of $\nabla_2^2 f(z^*)^2$ are the square of the eigenvalues of $\nabla_2^2 f(z^*)$ and the eigenvalues of $\nabla_2^2 f(z^*)$ are bounded between $-\mu$ and $-L_2$ by Lemma 8 and the Lipschitz assumption.

Then, with this choice of $P_2$, we have that

$$P_2(\nabla_2^2 f(z^*))^{-1} \nabla_{12}^\top f(z^*) \nabla_{12} f(z^*) + (P_2(\nabla_2^2 f(z^*))^{-1} \nabla_{12}^\top f(z^*) \nabla_{12} f(z^*))^\top = 2\nabla_{12}^\top f(z^*) \nabla_{12} f(z^*) \succeq 0.$$

Now, since $B_{\tau_0} \succ 0$ so is $Q_{\tau_0}$ since they are congruent. Since $\mathrm{spec}(-J_{\tau_0}(z^*)) \subset \mathbb{C}_-^\circ$, $Q_{\tau_0} \succ 0$ implies that $P = P^\top \prec 0$ (by Lyapunov's theorem). Hence, $P_1$ and $P_2$ must be negative definite since $P$ is congruent to $\mathrm{diag}(P_1, P_2)$, but this gives us a contradiction with the fact that $P_1$ has the same inertia as $-\mathtt{S}_1(J_1(z^*))$ which we assumed to have at least one positive eigenvalue. Hence, if $\mathrm{spec}(-J_{\tau_0}(z^*)) \subset \mathbb{C}_-^\circ$ for some $\tau_0 \in (0, \infty)$, then it must be the case that $\mathtt{S}_1(J_1(z^*)) \succ 0$ which means $z^*$ is a differential Stackelberg equilibrium since we also have $-\nabla_2^2 f(z^*) \succ 0$ by Lemma 8.

Thus, we can finish the proof of part 1 as follows. Consider any critical point $\tilde{z}$ that is not a differential Stackelberg equilibrium and is unstable for the nominal $\tau_0$. Then, we claim that $\mathrm{spec}(-J_\tau(\tilde{z})) \not\subset \mathbb{C}_-^\circ$ for all $\tau \geq \tau_0$. Suppose not. That is, there is some $\tau_1 \geq \tau_0$ such that $\mathrm{spec}(-J_{\tau_1}(\tilde{z})) \subset \mathbb{C}_-^\circ$. But by our argument above, since $-\nabla_2^2 f(\tilde{x}) \succ 0$, this implies that $\mathtt{S}_1(J_1(\tilde{z})) \succ 0$ which contradicts that $\tilde{z}$ is not a differential Stackelberg equilibrium. Hence, any critical point $z^*$ that is not a differential Stackelberg equilibrium is unstable for all $\tau \in (0, \infty)$.

**Proof of 2.** We note that the fact there exists a finite $\tau^* \in (0, \infty)$ such that a differential Stackelberg is stable is known [15]. Let $\tau^*$ denote the minimum $\tau^*$ such that a differential Stackelberg equilibrium $z^*$ is stable. To see that $\mathrm{spec}(-J_\tau(z^*)) \subset \mathbb{C}_-^\circ$ for all $\tau \geq \tau^*$ given that $\mathrm{spec}(-J_{\tau^*}(z^*)) \subset \mathbb{C}_-^\circ$, we can again examine the Lyapunov equation under the congruent transformation. We define the matrix $P$ as above in equation (11) where $P_1 \prec 0$ and $P_2 \prec 0$ since $-\mathtt{S}(J_1(z^*)) \prec 0$ and $\nabla_2^2 f(z^*) \prec 0$ with $Q_1, Q_2 \succ 0$. With

$$Q_\tau = \begin{bmatrix} I & L_0^\top \\ 0 & I \end{bmatrix} B_\tau \begin{bmatrix} I & 0 \\ L_0 & I \end{bmatrix}$$

and

$$B_\tau = \begin{bmatrix} Q_1 & P_1 \nabla_{12} f(z^*) - \mathtt{S}(J_1(z^*)) L_0^\top P_2 \\ (P_1 \nabla_{12} f(z^*) - \mathtt{S}(J_1(z^*)) L_0^\top P_2)^\top & P_2 L_0 \nabla_{12} f(z^*) + (P_2 L_0 \nabla_{12} f(z^*))^\top + \tau Q_2 \end{bmatrix}$$

we again can see that $Q_\tau \succ 0$ for the same reason as above for any $\tau \geq \tau^*$. This, in turn, implies that $\mathrm{spec}(-J_\tau(z^*)) \subset \mathbb{C}_-^\circ$ for all $\tau \geq \tau^*$ since we constructed a Lyapunov function for $z^*$. Thus, we conclude that if $z^*$ is a differential Stackelberg equilibrium, then $\mathrm{spec}(-J_\tau(z^*)) \subset \mathbb{C}_-^\circ$ for all $\tau \in [\tau_*, \infty)$ where $\tau_*$ is the minimum $\tau \in (0, \infty)$ such that $\mathrm{spec}(-J_\tau(z^*)) \subset \mathbb{C}_-^\circ$ and a finite $\tau_*$ is guaranteed to exist. $\qquad\square$

# D   Convergence Analysis for Nonconvex-PŁ Zero-Sum Games

We now provide the proofs pertaining to the results presented in Section 5 for nonconvex-PŁ zero-sum games. To help the presentation, Table 2 includes the relevant game and smoothness parameters needed in the proofs.

| Parameter | Meaning/Value |
|---|---|
| $\mu$ | $f(x, \cdot)$ is $\mu$-PŁ or $\mu$-SC for all $x \in \mathcal{X}$ |
| $L_1$ | $\forall\, x, x' \in \mathcal{X}, y, y' \in \mathcal{Y} : \|\nabla_1 f(x, y) - \nabla_1 f(x', y')\| \leq L_1(\|x - x'\| + \|y - y'\|)$ |
| $L_2$ | $\forall\, x, x' \in \mathcal{X}, y, y' \in \mathcal{Y} : \|\nabla_2 f(x, y) - \nabla_2 f(x', y')\| \leq L_2(\|x - x'\| + \|y - y'\|)$ |
| $L$ | $L = L_1 + L_2$ and $\forall\, x, x' \in \mathcal{X}, y, y' \in \mathcal{Y} : \|g(x, y) - g(x', y')\| \leq L(\|x - x'\| + \|y - y'\|)$ |
| $L_3$ | $L_3 = L_2/\mu$ and $\forall\, x, x' \in \mathcal{X} : \|y_*(x) - y_*(x')\| \leq L_3\|x - x'\|$ |
| $L_4$ | $L_4 = L_1 + \kappa L_2$ and $\forall\, x, x' \in \mathcal{X} : \|f(x, y_*(x)) - f(x', y_*(x'))\| \leq L_4\|x - x'\|$ |
| $\kappa$ | $L_1/\mu$ |
| $\kappa'$ | $L_2/\mu$ |
| $\ell$ | $L_1 + 3\kappa' L_2$ |

Table 2: Table of Lipschitz Parameters

### D.1 Proof of Lemma 1

This appendix is devoted to proving Lemma 1. To be clear, we restate the result before proving it.

**Lemma 1.** *Consider a nonconvex-PŁ/SC zero-sum game defined by $f \in C^2(\mathcal{Z}, \mathbb{R})$ satisfying Assumption 1. For any $\Gamma \in (0, 1/7]$, suppose that $\tau \geq \Gamma^{-1}\kappa^2$ and $\gamma < \min\{\frac{1}{2L}, \frac{1}{2\tau L}, \frac{1}{2L_4}\}$, then $\Phi(x, y) = f(x, y_*(x)) - \Gamma f(x, y)$ is a potential function for $\tau$-GDA.*

*Proof of Lemma 1.* Let the best response be denoted by

$$y_*(x) \in \operatorname*{argmax}_{y \in \mathcal{Y}} f(x, y).$$

Since the function $f(x, \cdot)$ is PŁ, the set maximizers may not be a singleton. Hence, $y_*(x)$ is an element of the set of maximizers. Recall that

$$\Phi(x, y) := f(x, y_*(x)) - \Gamma f(x, y).$$

We claim that for any $\Gamma \in (0, 1/7]$,

$$\Phi(x_{k+1}, y_{k+1}) - \Phi(x_k, y_k) = \Gamma \underbrace{(f(x_k, y_k) - f(x_{k+1}, y_{k+1}))}_{(i)} + \underbrace{f(x_{k+1}, y_*(x_{k+1})) - f(x_k, y_*(x_k))}_{(ii)} < 0.$$

To show this, we need to bound each of the two terms (i) and (ii).

**Bounding term (i): $f(x_k, y_k) - f(x_{k+1}, y_{k+1})$.** To begin, we add and subtract $f(x_k, y_{k+1})$ to (i) and get

$$f(x_k, y_k) - f(x_{k+1}, y_{k+1}) = f(x_k, y_k) - f(x_k, y_{k+1}) + f(x_k, y_{k+1}) - f(x_{k+1}, y_{k+1}). \tag{13}$$

From a Taylor expansion of $-f(x_k, y_{k+1})$ with respect to $y_{k+1}$ we obtain

$$-f(x_k, y_{k+1}) \leq -f(x_k, y_k) - \langle \nabla_2 f(x_k, y_k), y_{k+1} - y_k \rangle + \frac{L_2}{2}\|y_{k+1} - y_k\|^2$$

$$= -f(x_k, y_k) - \tau\gamma\langle \nabla_2 f(x_k, y_k), \nabla_2 f(x_k, y_k) \rangle + \frac{\tau^2\gamma^2 L_2}{2}\|\nabla_2 f(x_k, y_k)\|^2$$

$$= -f(x_k, y_k) - \left(\tau\gamma - \frac{\tau^2\gamma^2 L_2}{2}\right)\|\nabla_2 f(x_k, y_k)\|^2.$$

Hence, rearranging this bound and combining with (13), we get

$$f(x_k, y_k) - f(x_{k+1}, y_{k+1}) \leq -\left(\tau\gamma - \frac{\tau^2\gamma^2 L_2}{2}\right)\|\nabla_2 f(x_k, y_k)\|_2^2 + f(x_k, y_{k+1}) - f(x_{k+1}, y_{k+1}). \tag{14}$$

Now, from a Taylor expansion of $-f(x_{k+1}, y_{k+1})$ with respect to $x_{k+1}$, we get

$$-f(x_{k+1}, y_{k+1}) \leq -f(x_k, y_{k+1}) - \langle \nabla_1 f(x_k, y_{k+1}), x_{k+1} - x_k \rangle + \frac{L_1}{2}\|x_{k+1} - x_k\|^2$$

$$= -f(x_k, y_{k+1}) + \gamma\langle \nabla_1 f(x_k, y_{k+1}), \nabla_1 f(x_k, y_k) \rangle + \frac{L_1\gamma^2}{2}\|\nabla_1 f(x_k, y_k)\|^2.$$

We now add and subtract $\gamma\|\nabla_1 f(x_k, y_k)\|^2$ to obtain

$$f(x_k, y_{k+1}) - f(x_{k+1}, y_{k+1}) \leq \gamma\langle \nabla_1 f(x_k, y_{k+1}), \nabla_1 f(x_k, y_k) \rangle + \frac{L_1\gamma^2}{2}\|\nabla_1 f(x_k, y_k)\|^2$$

$$- \gamma\langle \nabla_1 f(x_k, y_k), \nabla_1 f(x_k, y_k) \rangle + \gamma\|\nabla_1 f(x_k, y_k)\|_2^2$$

$$= \gamma\langle \nabla_1 f(x_k, y_{k+1}) - \nabla_1 f(x_k, y_k), \nabla_1 f(x_k, y_k) \rangle + \frac{L_1\gamma^2 + 2\gamma}{2}\|\nabla_1 f(x_k, y_k)\|^2. \tag{15}$$

Next, we combine this with (14) to get

$$f(x_k, y_k) - f(x_{k+1}, y_{k+1}) \leq -\left(\tau\gamma - \frac{\tau^2\gamma^2 L_2}{2}\right)\|\nabla_2 f(x_k, y_k)\|^2$$

$$+ \gamma\langle \nabla_1 f(x_k, y_{k+1}) - \nabla_1 f(x_k, y_k), \nabla_1 f(x_k, y_k) \rangle + \frac{L_1\gamma^2 + 2\gamma}{2}\|\nabla_1 f(x_k, y_k)\|^2$$

$$\leq -\left(\tau\gamma - \frac{\tau^2\gamma^2 L_2}{2}\right)\|\nabla_2 f(x_k, y_k)\|^2 + \frac{L_1\gamma^2 + 2\gamma}{2}\|\nabla_1 f(x_k, y_k)\|^2$$

$$+ \gamma\|\nabla_1 f(x_k, y_{k+1}) - \nabla_1 f(x_k, y_k)\|\|\nabla_1 f(x_k, y_k)\|.$$

Note that the final inequality is a result of applying Cauchy-Schwartz. Applying Young's inequality on the last term in the inequality above, we have that

$$
\begin{aligned}
f(x_k, y_k) - f(x_{k+1}, y_{k+1}) &\leq -\left(\tau\gamma - \frac{\tau^2\gamma^2 L_2}{2}\right)\|\nabla_2 f(x_k, y_k)\|^2 + \frac{L_1\gamma^2 + 2\gamma + \gamma}{2}\|\nabla_1 f(x_k, y_k)\|^2 \\
&\quad + \frac{\gamma}{2}\|\nabla_1 f(x_k, y_{k+1}) - \nabla_1 f(x_k, y_k)\|^2 \\
&\leq -\left(\tau\gamma - \frac{\tau^2\gamma^2 L_2}{2} - \frac{\gamma(\tau\gamma)^2 L_1^2}{2}\right)\|\nabla_2 f(x_k, y_k)\|^2 + \frac{L_1\gamma^2 + 2\gamma + \gamma}{2}\|\nabla_1 f(x_k, y_k)\|^2.
\end{aligned}
$$
(16)

Observe that the final inequality is a result of applying the Lipschitz bound

$$
\frac{\gamma}{2}\|\nabla_1 f(x_k, y_{k+1}) - \nabla_1 f(x_k, y_k)\|^2 \leq \frac{\gamma L_1^2}{2}\|y_{k+1} - y_k\|^2 = \frac{\gamma(\tau^2\gamma^2)L_1^2}{2}\|\nabla_2 f(x_k, y_k)\|^2.
$$

**Bounding (ii):** $f(x_{k+1}, y_*(x_{k+1})) - f(x_k, y_*(x_k))$. To bound (ii), we take a Taylor expansion of $f(x_{k+1}, y_*(x_{k+1}))$ to get that

$$
\begin{aligned}
f(x_{k+1}, y_*(x_{k+1})) - f(x_k, y_*(x_k)) &\leq \langle \nabla f(x_k, y_*(x_k)), x_{k+1} - x_k \rangle + \frac{L_4}{2}\|x_{k+1} - x_k\|^2 \\
&= -\gamma\langle \nabla f(x_k, y_*(x_k)), \nabla_1 f(x_k, y_k)\rangle + \frac{L_4\gamma^2}{2}\|\nabla_1 f(x_k, y_k)\|^2
\end{aligned}
$$
(17)

where $L_4$ is the gradient Lipschitz bound on the total derivative of $f(x, y_*(x))$ from Lemma 7.

**Combining bounds.** We now combine the bounds on (i) and (ii) from (16) and (17) to give that

$$
\begin{aligned}
\Phi(x_{k+1}, y_{k+1}) - \Phi(x_k, y_k) &= \Gamma \underbrace{(f(x_k, y_k) - f(x_{k+1}, y_{k+1}))}_{(i)} + \underbrace{f(x_{k+1}, y_*(x_{k+1})) - f(x_k, y_*(x_k))}_{(ii)} \\
&\leq -\Gamma\left(\tau\gamma - \frac{\tau^2\gamma^2 L_2}{2} - \frac{\gamma(\tau\gamma)^2 L_1^2}{2}\right)\|\nabla_2 f(x_k, y_k)\|^2 + \Gamma\left(\frac{L_1\gamma^2 + 2\gamma + \gamma}{2}\right)\|\nabla_1 f(x_k, y_k)\|^2 \\
&\quad - \gamma\langle\nabla f(x_k, y_*(x_k)), \nabla_1 f(x_k, y_k)\rangle + \frac{L_4\gamma^2}{2}\|\nabla_1 f(x_k, y_k)\|^2 \\
&= -\gamma\langle\nabla f(x_k, y_*(x_k)), \nabla_1 f(x_k, y_k)\rangle + \frac{\gamma}{2}(\Gamma L_1\gamma + 3\Gamma + L_4\gamma)\|\nabla_1 f(x_k, y_k)\|^2 \\
&\quad - \Gamma\left(\tau\gamma - \frac{\tau^2\gamma^2 L_2}{2} - \frac{\gamma(\tau\gamma)^2 L_1^2}{2}\right)\|\nabla_2 f(x_k, y_k)\|^2.
\end{aligned}
$$
(18)

To further bound the above expression, we start by bounding the first two terms. Towards this end, define

$$
V := -\gamma\langle\nabla f(x_k, y_*(x_k)), \nabla_1 f(x_k, y_k)\rangle + \frac{\gamma}{2}(\Gamma L_1\gamma + 3\Gamma + L_4\gamma)\|\nabla_1 f(x_k, y_k)\|^2.
$$
(19)

Recall that $\gamma < 1/(2L_4)$ and $\gamma < 1/(2L)$ and observe that this implies $\gamma < 1/(2L_1)$ since $L = L_1 + L_2 > L_1$. Thus, since $\Gamma \leq 1/7$,

$$
\gamma\Gamma L_1 + 3\Gamma + \gamma L_4 \leq \frac{\Gamma}{2} + 3\Gamma + \frac{1}{2} = \frac{1}{2} + \frac{7\Gamma}{2} \leq 1.
$$

Thus, applying this fact, we can bound (19) as follows:

$$
V \leq -\gamma\langle\nabla f(x_k, y_*(x_k)), \nabla_1 f(x_k, y_k)\rangle + \frac{\gamma}{2}\|\nabla_1 f(x_k, y_k)\|^2.
$$
(20)

Now, by adding and subtracting $\frac{\gamma}{2}\|\nabla f(x_k, y_*(x_k))\|^2$ in (20) and then simplifying, we have that

$$
\begin{aligned}
V &\leq -\frac{\gamma}{2}\|\nabla f(x_k, y_*(x_k))\|^2 + \frac{\gamma}{2}\|\nabla f(x_k, y_*(x_k))\|^2 - \gamma\langle\nabla f(x_k, y_*(x_k)), \nabla_1 f(x_k, y_k)\rangle + \frac{\gamma}{2}\|\nabla_1 f(x_k, y_k)\|^2 \\
&= -\frac{\gamma}{2}\|\nabla f(x_k, y_*(x_k))\|^2 + \frac{\gamma}{2}\left(\|\nabla f(x_k, y_*(x_k))\|^2 - 2\langle\nabla f(x_k, y_*(x_k)), \nabla_1 f(x_k, y_k)\rangle + \|\nabla_1 f(x_k, y_k)\|^2\right) \\
&= -\frac{\gamma}{2}\|\nabla f(x_k, y_*(x_k))\|^2 + \frac{\gamma}{2}\|\nabla f(x_k, y_*(x_k)) - \nabla_1 f(x_k, y_k)\|^2.
\end{aligned}
$$
(21)

Moreover, we can continue to bound (21) in the following way that is explained below:

$$V \leq -\frac{\gamma}{2}\|\nabla f(x_k, y_*(x_k))\|^2 + \frac{\gamma}{2}\|\nabla_1 f(x_k, y_*(x_k)) - \nabla_1 f(x_k, y_k)\|^2 \tag{22}$$

$$\leq -\frac{\gamma}{2}\|\nabla f(x_k, y_*(x_k))\|^2 + \frac{\gamma L_1^2}{2}\|y_*(x_k) - y_k\|^2 \tag{23}$$

$$\leq -\frac{\gamma}{2}\|\nabla f(x_k, y_*(x_k))\|^2 + \frac{\gamma \kappa^2}{2}\|\nabla_2 f(x_k, y_k)\|^2. \tag{24}$$

Observe that (22) is a result of applying the fact $\nabla f(x_k, y_*(x_k)) = \nabla_1 f(x, y)|_{y=y_*(x)}$ by Lemma 7 and gives an equivalent formulation of (21). Then in (23) we used that $\nabla_1 f(x, y)$ is $L_1$-Lipschitz in $y$ by Assumption 1. Furthermore, in (24), we used the error bound property of $\mu$-PŁ functions from Lemma 5 to get that

$$\|y_*(x_k) - y_k\|^2 \leq \frac{1}{\mu^2}\|\nabla_2 f(x_k, y_k)\|^2.$$

Finally, also in (24), we applied the condition number definition $\kappa = L_1/\mu$.

Now, by combining the bound on $V$ from (24) with the remaining terms in (18), we have

$$\Phi(x_{k+1}, y_{k+1}) - \Phi(x_k, y_k) \leq -\frac{\gamma}{2}\|\nabla f(x_k, y_*(x_k))\|^2 + \frac{\gamma \kappa^2}{2}\|\nabla_2 f(x_k, y_k)\|^2$$

$$- \Gamma\Big(\tau\gamma - \frac{\tau^2\gamma^2 L_2}{2} - \frac{\gamma(\tau\gamma)^2 L_1^2}{2}\Big)\|\nabla_2 f(x_k, y_k)\|^2$$

$$= -\frac{\gamma}{2}\|\nabla f(x_k, y_*(x_k))\|^2 + \frac{\tau\gamma\Gamma}{2}\Big(\frac{\kappa^2}{\tau\Gamma} + \tau\gamma L_2 + \tau\gamma^2 L_1^2 - 2\Big)\|\nabla_2 f(x_k, y_k)\|^2. \tag{25}$$

Let

$$C := \frac{\kappa^2}{\tau\Gamma} + \tau\gamma L_2 + \tau\gamma^2 L_1^2 - 2.$$

As long as $C < 0$, then the function $\Phi(\cdot, \cdot)$ is decreasing along the trajectories of $\tau$-GDA. To see this, we upper bound $C$ in the following manner that is explained below:

$$C = \frac{\kappa^2}{\tau\Gamma} + \tau\gamma L_2 + \tau\gamma^2 L_1^2 - 2$$

$$\leq \tau\gamma L_2 + \tau\gamma^2 L_1^2 - 1 \tag{26}$$

$$\leq \frac{\gamma L_1}{2} - \frac{1}{2} \tag{27}$$

$$\leq -1/4. \tag{28}$$

The inequality in (26) is obtained using the fact that $\Gamma^{-1}\kappa^2 \leq \tau$. Moreover, (27) follows from the fact that $\gamma < 1/(2\tau L) < \min\{1/(2\tau L_1), 1/(2\tau L_2)\}$. Finally, (28) holds since $\gamma < 1/(2L) < \min\{1/(2L_1), 1/(2L_2)\}$.

Hence, combining (25) and (28) gives

$$\Phi(x_{k+1}, y_{k+1}) - \Phi(x_k, y_k) \leq -\frac{\gamma}{2}\|\nabla f(x_k, y_*(x_k))\|^2 - \frac{\tau\gamma\Gamma}{8}\|\nabla_2 f(x_k, y_k)\|^2.$$

Thus, the function $\Phi(\cdot, \cdot)$ only stops decreasing when we have both

$$\|\nabla f(x_k, y_*(x_k))\|^2 = 0 \quad \text{and} \quad \|\nabla_2 f(x_k, y_k)\|^2 = 0.$$

By the error bound property of $\mu$-PŁ functions from Lemma 5 we have

$$\|y_*(x_k) - y_k\|^2 \leq \frac{1}{\mu^2}\|\nabla_2 f(x_k, y_k)\|^2.$$

Hence, if $\|\nabla_2 f(x_k, y_k)\|^2 \to 0$ then $y_k \to y_*(x_k)$. In particular, when $\|\nabla_2 f(x_k, y_k)\|^2 = 0$, we have that $y_k = y_*(x_k)$ so that $\|\nabla f(x_k, y_*(x_k))\|^2 = 0$ if and only if $\|\nabla_1 f(x_k, y_k)\|^2 = 0$. This implies that $\Phi(\cdot, \cdot)$ only stops decreasing along the $\tau$-GDA iterates at critical points of $\tau$-GDA. Moreover, observe that $\Phi(x, y) \geq 0$ for any $(x, y) \in \mathcal{X} \times \mathcal{Y}$ since by definition of $y_*(x)$ it immediately follows that $f(x, y_*(x)) \geq f(x, y)$ so that $f(x, y_*(x)) - \Gamma f(x, y) \geq 0$ owing to the fact that $\Gamma \in (0, 1/7]$. Thus, since $\Phi(\cdot, \cdot)$ is bounded below, it must stop decreasing along the $\tau$-GDA iterates. Thus, $\Phi(\cdot, \cdot)$ is a potential function as claimed. $\qquad\square$

## D.2 Proof of Theorem 2

In this appendix, we prove Theorem 2. To be clear, we restate the result before proving it.

**Theorem 2.** *Consider a nonconvex-PŁ/SC zero-sum game $(f, -f)$ defined by $f \in C^2(\mathcal{Z}, \mathbb{R})$ that satisfies Assumptions 1–2. Then, $\tau$-GDA with $\tau \geq \Gamma^{-1}\kappa^2$, and $\gamma < \min\{\frac{1}{2L}, \frac{1}{2\tau L}, \frac{1}{2L_4}\}$ for $\Gamma \in (0, 1/7]$ asymptotically converges to the set of strict local minmax that are stable for $\dot{z} = -\Lambda_\tau g(z)$ almost surely. That is, for almost all initial conditions, $\tau$-GDA will converge to a strict local minmax.*

*Proof of Theorem 2.* This result follows nearly immediately from Theorem 1, Lemma 1, and Lemma 2. In particular, the potential function result from Lemma 1 guarantees that $\tau$-GDA converges to a critical (stationary) point of the update for the choice of $\gamma$ and $\tau$. Then, since the only stable points of the $\tau$-GDA continuous-time $\dot{z} = -\Lambda_\tau g(z)$ dynamics are strict local minmax (differential Stackelberg) equilibrium by Theorem 1, $\tau$-GDA avoids strict saddle points of $\dot{z} = -\Lambda_\tau g(z)$ almost surely for the choice of $\gamma$ and $\tau$ by Lemma 2, and all saddle points of $\dot{z} = -\Lambda_\tau g(z)$ are assumed to be strict saddle points by Assumption 2 for the given choice of $\tau$, we can conclude that $\tau$-GDA almost surely converges to a strict local minmax (differential Stackelberg) equilibrium. $\square$

## D.3 Proof of Corollary 1

In this appendix, we prove Corollary 1. We restate the result and then provide the proof.

**Corollary 1.** *Consider a nonconvex-PŁ/SC zero-sum game $(f, -f)$ defined by $f \in C^2(\mathcal{Z}, \mathbb{R})$ that satisfying Assumption 1. Then, $\tau$-GDA from any initialization with $\tau \geq \Gamma^{-1}\kappa^2$, $\gamma < \min\{\frac{1}{2L}, \frac{1}{2\tau L}, \frac{1}{2L_4}\}$ for $\Gamma \in (0, 1/7]$, has at least one iterate that is an $\varepsilon$-critical point after $\widetilde{\mathcal{O}}(\varepsilon^{-2})$ iterations.*

*Proof of Corollary 1.* For this proof, consider any $\varepsilon > 0$. Our approach will be to show that for

$$T \geq \frac{2\Phi(x_0, y_0)}{\varepsilon^2 \gamma \min\{1, \tau\Gamma/4\}},$$

we have that

$$\min_{0 \leq k \leq T-1} \max\left\{\|\nabla f(x_k, y_*(x_k))\|, \|\nabla_2 f(x_k, y_k)\|\right\} := \max\left\{\|\nabla f(x_s, y_*(x_s))\|, \|\nabla_2 f(x_s, y_s)\|\right\} \leq \varepsilon.$$

Then, we prove that given this fact

$$\|\nabla_1 f(x_s, y_s)\| \leq \left(1 + \frac{L_1}{\mu}\right)\varepsilon.$$

This will then allow us to conclude for

$$T \geq \frac{2\left(1 + \frac{L_1}{\mu}\right)^2 \Phi(x_0, y_0)}{\varepsilon^2 \gamma \min\{1, \tau\Gamma/4\}}, \tag{29}$$

we have both

$$\|\nabla_1 f(x_s, y_s)\| \leq \varepsilon \quad \text{and} \quad \|\nabla_2 f(x_s, y_s)\| \leq \varepsilon.$$

Then, by selecting the parameters to minimize the right-hand side of (29), we are able to conclude that at least one iterate of the $\tau$-GDA dynamics are an $\varepsilon$-critical point after

$$T \geq \frac{2\left(1 + \frac{L_1}{\mu}\right)^2 \Phi(x_0, y_0)}{\varepsilon^2 \gamma \min\{1, \tau\Gamma/4\}}$$

iterations.

We now formally prove this. Summing the bound on the potential function from Lemma 1, we get the following that is justified below:

$$\Phi(x_0, y_0) \geq \Phi(x_0, y_0) - \Phi(x_T, y_T) \tag{30}$$

$$= \sum_{k=0}^{T-1} \left( \Phi(x_k, y_k) - \Phi(x_{k+1}, y_{k+1}) \right) \tag{31}$$

$$\geq \frac{\gamma}{2} \sum_{k=0}^{T-1} \|\nabla f(x_k, y_*(x_k))\|^2 + \frac{\tau\gamma\Gamma}{8} \sum_{k=0}^{T-1} \|\nabla_2 f(x_k, y_k)\|^2 \tag{32}$$

$$\geq \frac{\gamma}{2} \min\left\{1, \frac{\tau\Gamma}{4}\right\} \sum_{k=0}^{T-1} \left( \|\nabla f(x_k, y_*(x_k))\|^2 + \|\nabla_2 f(x_k, y_k)\|^2 \right) \tag{33}$$

$$\geq \frac{\gamma}{2} \min\left\{1, \frac{\tau\Gamma}{4}\right\} \sum_{k=0}^{T-1} \max\left\{ \|\nabla f(x_k, y_*(x_k))\|^2, \|\nabla_2 f(x_k, y_k)\|^2 \right\} \tag{34}$$

$$\geq \frac{\gamma T}{2} \min\left\{1, \frac{\tau\Gamma}{4}\right\} \min_{0 \leq k \leq T-1} \max\left\{ \|\nabla f(x_k, y_*(x_k))\|^2, \|\nabla_2 f(x_k, y_k)\|^2 \right\}. \tag{35}$$

Note that (30) holds since $\Phi(x, y) \geq 0$ for any $(x, y) \in \mathcal{X} \times \mathcal{Y}$ since by definition of $y_*(x)$ it immediately follows that $f(x, y_*(x)) \geq f(x, y)$ so that $f(x, y_*(x)) - \Gamma f(x, y) \geq 0$ owing to the fact that $\Gamma \in (0, 1/7]$. Observe that (31) follows from telescoping of the sum, (32) is a result of applying the bound on the potential function, (33) holds since it is replacing a coefficient of a positive number with something smaller, (34) holds since the sum of positive numbers is greater than the max, and (35) is obtained from the fact that the sum of $T$ positive numbers is greater than $T$ times the minimum number.

From the previous steps, and also rearranging terms and then taking the square root, we have

$$\sqrt{\frac{2\Phi(x_0, y_0)}{T\gamma \min\{1, \tau\Gamma/4\}}} \geq \min_{0 \leq k \leq T-1} \max\left\{ \|\nabla f(x_k, y_*(x_k))\|, \|\nabla_2 f(x_k, y_k)\| \right\}.$$

We now want to find $T$ such that

$$\min_{0 \leq k \leq T-1} \max\left\{ \|\nabla f(x_k, y_*(x_k))\|, \|\nabla_2 f(x_k, y_k)\| \right\} \leq \sqrt{\frac{2\Phi(x_0, y_0)}{T\gamma \min\{1, \tau\Gamma/4\}}} \leq \varepsilon. \tag{36}$$

By moving terms around, we find that the inequality in (36) holds for any $T$ such that

$$T \geq T^* := \frac{2\Phi(x_0, y_0)}{\varepsilon^2 \gamma \min\{1, \tau\Gamma/4\}}. \tag{37}$$

This proves that there exists some iterate $0 \leq s \leq T-1$ such that for $T \geq T^*$, we have both

$$\|\nabla f(x_s, y_*(x_s))\| \leq \varepsilon \quad \text{and} \quad \|\nabla_2 f(x_s, y_s)\| \leq \varepsilon.$$

We now show that this implies a bound on $\|\nabla_1 f(x_s, y_s)\|$. In particular, by the error bound property of $\mu$-PŁ functions from Lemma 5 we have

$$\|y_s - y_*(x_s)\|^2 \leq \frac{1}{\mu^2} \|\nabla_2 f(x_s, y_s)\|^2.$$

Since $\|\nabla_2 f(x_s, y_s)\| \leq \varepsilon$, we know that $\|y_s - y_*(x_s)\| \leq \frac{\varepsilon}{\mu}$.

Then, observe that we have the following bound explained below

$$\|\nabla_1 f(x_s, y_s)\| = \|\nabla_1 f(x_s, y_s) - \nabla f(x_s, y_*(x_s)) + \nabla f(x_s, y_*(x_s))\|$$

$$\leq \|\nabla_1 f(x_s, y_s) - \nabla f(x_s, y_*(x_s))\| + \|\nabla f(x_s, y_*(x_s))\| \tag{38}$$

$$= \|\nabla_1 f(x_s, y_s) - \nabla_1 f(x_s, y_*(x_s))\| + \|\nabla f(x_s, y_*(x_s))\| \tag{39}$$

$$\leq L_1 \|y_s - y_*(x_s)\| + \|\nabla f(x_s, y_*(x_s))\| \tag{40}$$

$$\leq \frac{L_1 \varepsilon}{\mu} + \varepsilon = \left(1 + \frac{L_1}{\mu}\right)\varepsilon. \tag{41}$$

Observe that (38) follows from the triangle inequality, (39) is a result of applying the fact $\nabla f(x_s, y_*(x_s)) = \nabla_1 f(x_s, y)|_{y=y_*(x_s)}$ by Lemma 7, in (23) we used that $\nabla_1 f(x, y)$ is $L_1$-Lipschitz in $y$ by Assumption 1, and (41) applies the inequality above that $\|y_s - y_*(x_s)\| \leq \frac{\varepsilon}{\mu}$.

Thus, in order to determine the iteration complexity $T$ needed to get that $\|\nabla_1 f(x_s, y_s)\| \leq \varepsilon$, we can consider the $T^*$ that ensures there exists some iterate $0 \leq s \leq T - 1$ such that for $T \geq T^*$, we have both

$$\|\nabla f(x_s, y_*(x_s))\| \leq \varepsilon \left(1 + \frac{L_1}{\mu}\right)^{-1} \quad \text{and} \quad \|\nabla_2 f(x_s, y_s)\| \leq \varepsilon \left(1 + \frac{L_1}{\mu}\right)^{-1}.$$

This amounts to replacing $\varepsilon$ in (37) with $\varepsilon(1 + \frac{L_1}{\mu})^{-1}$ to get that for

$$T \geq T^* := \frac{2\left(1 + \frac{L_1}{\mu}\right)^2 \Phi(x_0, y_0)}{\varepsilon^2 \gamma \min\{1, \tau\Gamma/4\}}, \tag{42}$$

we have both

$$\|\nabla_1 f(x_s, y_s)\| \leq \varepsilon \quad \text{and} \quad \|\nabla_2 f(x_s, y_s)\| \leq \varepsilon.$$

This holds for $\tau > 7\kappa^2$, $\gamma < \min\{\frac{1}{2L}, \frac{1}{2\tau L}, \frac{1}{2L_4}\}$, and any $\Gamma \in (0, 1/7]$ by Lemma 1. Thus, we can see that the complexity is $\widetilde{\mathcal{O}}(\varepsilon^{-2})$ to reach an $\varepsilon$-critical point under the given parameter choices. To obtain an explicit bound for fixed choices of $\tau$, $\gamma$, and $\Gamma$, we can select the parameters to minimize the right hand side of (42). In particular, selecting $\tau = 8\kappa^2$, $\Gamma = 1/8$, and $\gamma = \frac{1}{2}\min\{\frac{1}{2L}, \frac{1}{2\tau L}, \frac{1}{2L_4}\}$ gives an iteration complexity of

$$T^* = \frac{32\left(1 + \frac{L_1}{\mu}\right)^2 \max\{L, 8\kappa^2 L, L_4\}\Phi(x_0, y_0)}{\varepsilon^2 \min\{4, \kappa^2\}}.$$

This completes the proof. $\qquad\qquad\square$

# E  Convergence Analysis for Nonconvex-SC Zero-Sum Games

This appendix contains the analysis for the results from Section 5.2 on global convergence guarantees to $\varepsilon$-critical points in nonconvex-SC zero-sum games. We present proofs for the stochastic descent in Appendix E.1, and convergence in Appendix E.2. Before proceeding, we restate the stochastic $\tau$-GDA update rule from Algorithm 2 and also recall relevant notation.

**Stochastic $\tau$–GDA Dynamics.**  Recall from Algorithm 2 that the combined update at each time $k$ of the $\tau$-GDA dynamics is given by

$$\begin{aligned}
x_{k+1} &= x_k - \gamma g_1(x_k, y_k; \theta_{1,k}) = x_k - \gamma(\nabla_1 f(x_k, y_k) + \zeta_{1,k}) \\
y_{k+1} &= y_k + \gamma\tau g_2(x_k, y_k; \theta_{2,k}) = y_k + \gamma\tau(\nabla_2 f(x_k, y_k) + \zeta_{2,k}),
\end{aligned}$$

so that

$$\begin{aligned}
x_{k+1} - x_k &= -\gamma g_1(x_k, y_k; \theta_{1,k}) = -\gamma(\nabla_1 f(x_k, y_k) + \zeta_{1,k}) \\
y_{k+1} - y_k &= \gamma\tau g_2(x_k, y_k; \theta_{2,k}) = \gamma\tau(\nabla_2 f(x_k, y_k) + \zeta_{2,k}).
\end{aligned}$$

In particular, $g_i(x_k, y_k; \theta_{i,k})$ denotes the stochastic gradient at step $k$ for player $i \in \mathcal{I} = \{1, 2\}$ with $\theta_{i,k}$ a random variable drawn from the distribution $\mathcal{D}_i$ such that (by Assumption 3)

$$\mathbb{E}_{\theta_{i,k} \sim \mathcal{D}_i}[g_i(x_k, y_k; \theta_{i,k})] = \nabla_i f(x_k, y_k),$$

and for all $t \in \mathbb{R}$,

$$\mathbb{P}(\|g_i(x_k, y_k; \theta_{i,k}) - \nabla_i f(x_k, y_k)\| \geq t) \leq 2\exp\left(-\frac{t^2}{2\sigma_i^2}\right).$$

That is, the stochastic gradient for each player $i \in \mathcal{I}$ is $\sigma_i$-norm-subGaussian (see Definition 8). Moreover, for each player $i \in \mathcal{I}$, the noise in the stochastic gradient is denoted by

$$\zeta_{i,k} := g_i(x_k, y_k; \theta_{i,k}) - \nabla_i f(x_k, y_k).$$

The noise in the stochastic gradient denoted by $\zeta_{i,k}$ is itself a $\sigma_i$-norm-subGaussian random vector for each player $i \in \mathcal{I}$. This observation follows immediately from the fact that the stochastic gradient $g_i(x_k, y_k; \theta_{i,k})$ is $\sigma_i$-norm-subGaussian for each player $i \in \mathcal{I}$ and from the definition of a norm-subGaussian random vector (refer to Definition 8).

## E.1 Proof of Lemma 3

In this appendix, we prove Lemma 3. We restate it here more precisely. The decent lemma shows that for a nonconvex-SC zero-sum game, the function $f$ acts as a potential function for the $\tau$-GDA dynamics. In particular, we show that the function $f$ can be decomposed into a component that is decreasing along trajectories of $\tau$-GDA and a component that exhibits a possible increase due to randomness in the stochastic gradients. The primary reason for the decrease is large gradients in addition to the structure of the maximizing player's problem and the timescale separation which we exploit in the proof to ensure sufficient decrease of $f$ along trajectories. Recall that $\ell := L_1 + 3\kappa' L_2$ where $\kappa' := \frac{L_2}{\mu}$.

**Lemma 3** (Restatement of Lemma 3). *Consider a nonconvex-SC zero-sum game $(f, -f)$ defined by $f \in C^2(\mathcal{Z}, \mathbb{R})$ satisfying Assumption 1 and Assumption 3. There exists a absolute constant $c$ such that with probability at least $1 - 3e^{-\iota}$ for any $\iota > 0$, the sequence generated by (stochastic) $\tau$-GDA with parameters $\gamma \leq 1/\ell$ and $\tau = \Gamma/(\mu\gamma)$ for an absolute constant $\Gamma \geq 4$ satisfies*

$$f(x_k, y_k) - f(x_0, y_0) \leq -\frac{\gamma}{8} \sum_{t=0}^{k-1} \left( \|\nabla_1 f(x_t, y_t)\|^2 + \tau \|\nabla_2 f(x_t, y_t)\|^2 \right) + c\gamma \left( \sigma_1^2(\gamma \ell k + \iota) + \tau \gamma \sigma_2^2 \iota \right).$$

*Proof of Lemma 3.* We first argue a one step descent bound. Towards this end, take the Taylor expansion of $f(\cdot, y_k)$ to get that

$$
\begin{aligned}
f(x_{k+1}, y_k) - f(x_k, y_k) &\leq \langle \nabla_1 f(x_k, y_k), x_{k+1} - x_k \rangle + \frac{L_1}{2} \|x_{k+1} - x_k\|^2 \\
&= -\gamma \langle \nabla_1 f(x_k, y_k), \nabla_1 f(x_k, y_k) + \zeta_{1,k} \rangle + \frac{\gamma^2 L_1}{2} \|\nabla_1 f(x_k, y_k) + \zeta_{1,k}\|^2 \\
&\leq -\gamma \langle \nabla_1 f(x_k, y_k), \nabla_1 f(x_k, y_k) + \zeta_{1,k} \rangle + \frac{\gamma^2 L_1}{2} \left( \frac{3}{2} \|\nabla_1 f(x_k, y_k)\|^2 + 3\|\zeta_{1,k}\|^2 \right) \\
&\qquad\qquad\qquad\qquad\qquad\qquad\qquad\qquad\qquad\qquad\qquad\qquad\qquad\qquad\qquad\qquad\quad (43) \\
&= -\left( \gamma - \frac{3\gamma^2 L_1}{4} \right) \|\nabla_1 f(x_k, y_k)\|^2 - \gamma \langle \nabla_1 f(x_k, y_k), \zeta_{1,k} \rangle + \frac{3\gamma^2 L_1}{2} \|\zeta_{1,k}\|^2. \quad (44)
\end{aligned}
$$

Note that (43) follows from the Cauchy-Schwarz inequality and Young's inequality for products. Specifically, for any $\epsilon > 0$, we have

$$
\begin{aligned}
\|\nabla_1 f(x_k, y_k) + \zeta_{1,k}\|^2 &= \|\nabla_1 f(x_k, y_k)\|^2 + 2\langle \nabla_1 f(x_k, y_k), \zeta_{1,k} \rangle + \|\zeta_{1,k}\|^2 \\
&\leq \|\nabla_1 f(x_k, y_k)\|^2 + 2\|\nabla_1 f(x_k, y_k)\| \|\zeta_{1,k}\| + \|\zeta_{1,k}\|^2 \\
&\leq \|\nabla_1 f(x_k, y_k)\|^2 + 2\left( \frac{1}{2\epsilon} \|\nabla_1 f(x_k, y_k)\|^2 + \frac{\epsilon}{2} \|\zeta_{1,k}\|^2 \right) + \|\zeta_{1,k}\|^2 \\
&\leq \frac{1+\epsilon}{\epsilon} \|\nabla_1 f(x_k, y_k)\|^2 + (1+\epsilon) \|\zeta_{1,k}\|^2. \quad (45)
\end{aligned}
$$

Thus, selecting $\epsilon = 2$ gives rise to the inequality in (43).

Now, since $f(x_{k+1}, y)$ is $\mu$-strongly concave with respect to $y$, we have

$$f(x_{k+1}, y_{k+1}) - f(x_{k+1}, y_k) \leq \langle \nabla_2 f(x_{k+1}, y_k), y_{k+1} - y_k \rangle - \frac{\mu}{2} \|y_{k+1} - y_k\|^2. \quad (46)$$

We now add and subtract $\langle \nabla_2 f(x_k, y_k), y_{k+1} - y_k \rangle$ and then apply Cauchy-Schwarz to get that

$$
\begin{aligned}
f(x_{k+1}, y_{k+1}) - f(x_{k+1}, y_k) &\leq \langle \nabla_2 f(x_{k+1}, y_k), y_{k+1} - y_k \rangle - \frac{\mu}{2} \|y_{k+1} - y_k\|^2 \pm \langle \nabla_2 f(x_k, y_k), y_{k+1} - y_k \rangle \\
&= \langle \nabla_2 f(x_{k+1}, y_k) - \nabla_2 f(x_k, y_k), y_{k+1} - y_k \rangle - \frac{\mu}{2} \|y_{k+1} - y_k\|^2 \\
&\qquad + \langle \nabla_2 f(x_k, y_k), y_{k+1} - y_k \rangle \\
&\leq \|\nabla_2 f(x_{k+1}, y_k) - \nabla_2 f(x_k, y_k)\| \|y_{k+1} - y_k\| - \frac{\mu}{2} \|y_{k+1} - y_k\|^2 \\
&\qquad + \langle \nabla_2 f(x_k, y_k), y_{k+1} - y_k \rangle.
\end{aligned}
$$

By Young's inequality with $\epsilon > 0$ applied to the term $\|\nabla_2 f(x_{k+1}, y_k) - \nabla_2 f(x_k, y_k)\|\|y_{k+1} - y_k\|$, the fact that $\nabla_2 f(x, y)$ is $L_2$-Lipschitz in $x$ by Assumption 1, and the update equation for the $y$-player, we have that

$$
\begin{aligned}
f(x_{k+1}, y_{k+1}) - f(x_{k+1}, y_k) &\le \frac{1}{2\epsilon}\|\nabla_2 f(x_{k+1}, y_k) - \nabla_2 f(x_k, y_k)\|^2 + \frac{\epsilon}{2}\|y_{k+1} - y_k\|^2 - \frac{\mu}{2}\|y_{k+1} - y_k\|^2 \\
&\quad + \langle \nabla_2 f(x_k, y_k), y_{k+1} - y_k \rangle \\
&\le \frac{L_2^2}{2\epsilon}\|x_{k+1} - x_k\|^2 - \frac{\mu - \epsilon}{2}\|y_{k+1} - y_k\|^2 + \langle \nabla_2 f(x_k, y_k), y_{k+1} - y_k \rangle \\
&= \frac{\gamma^2 L_2^2}{2\epsilon}\|\nabla_1 f(x_k, y_k) + \zeta_{1,k}\|^2 - \frac{(\mu - \epsilon)(\tau\gamma)^2}{2}\|\nabla_2 f(x_k, y_k) + \zeta_{2,k}\|^2 \\
&\quad + \tau\gamma\langle \nabla_2 f(x_k, y_k), \nabla_2 f(x_k, y_k) + \zeta_{2,k} \rangle \\
&\le \frac{3\gamma^2 L_2^2}{4\epsilon}\|\nabla_1 f(x_k, y_k)\|^2 + \frac{3\gamma^2 L_2^2}{2\epsilon}\|\zeta_{1,k}\|^2 - \frac{(\mu - \epsilon)(\tau\gamma)^2}{2}\|\nabla_2 f(x_k, y_k) + \zeta_{2,k}\|^2 \\
&\quad + \tau\gamma\|\nabla_2 f(x_k, y_k)\|^2 + \tau\gamma\langle \nabla_2 f(x_k, y_k), \zeta_{2,k} \rangle. \qquad (47)
\end{aligned}
$$

Observe that the final inequality above is a result of applying the estimate

$$
\|\nabla_1 f(x_k, y_k) + \zeta_{1,k}\|^2 \le \frac{3}{2}\|\nabla_1 f(x_k, y_k)\|^2 + 3\|\zeta_{1,k}\|^2,
$$

which follows from the Cauchy-Schwarz inequality and Young's inequality for products as shown in (45).

To simplify the bound on $f(x_{k+1}, y_{k+1}) - f(x_{k+1}, y_k)$ given in (47), we expand the term on the right-hand side with $\|\nabla_2 f(x_k, y_k) + \zeta_{2,k}\|^2$ and then group common terms as follows:

$$
\begin{aligned}
f(x_{k+1}, y_{k+1}) - f(x_{k+1}, y_k) &\le \frac{3\gamma^2 L_2^2}{4\epsilon}\|\nabla_1 f(x_k, y_k)\|^2 + \frac{3\gamma^2 L_2^2}{2\epsilon}\|\zeta_{1,k}\|^2 - \frac{(\mu - \epsilon)(\tau\gamma)^2}{2}\|\nabla_2 f(x_k, y_k) + \zeta_{2,k}\|^2 \\
&\quad + \tau\gamma\|\nabla_2 f(x_k, y_k)\|^2 + \tau\gamma\langle \nabla_2 f(x_k, y_k), \zeta_{2,k} \rangle \\
&= \frac{3\gamma^2 L_2^2}{4\epsilon}\|\nabla_1 f(x_k, y_k)\|^2 + \frac{3\gamma^2 L_2^2}{2\epsilon}\|\zeta_{1,k}\|^2 + \tau\gamma\|\nabla_2 f(x_k, y_k)\|^2 + \tau\gamma\langle \nabla_2 f(x_k, y_k), \zeta_{2,k} \rangle \\
&\quad - \frac{(\mu - \epsilon)(\tau\gamma)^2}{2}\Big(\|\nabla_2 f(x_k, y_k)\|^2 + 2\langle \nabla_2 f(x_k, y_k), \zeta_{2,k} \rangle + \|\zeta_{2,k}\|^2\Big) \\
&= \frac{3\gamma^2 L_2^2}{4\epsilon}\|\nabla_1 f(x_k, y_k)\|^2 + \frac{3\gamma^2 L_2^2}{2\epsilon}\|\zeta_{1,k}\|^2 + \Big(\tau\gamma - \frac{(\mu - \epsilon)(\tau\gamma)^2}{2}\Big)\|\nabla_2 f(x_k, y_k)\|^2 \\
&\quad + \Big(\tau\gamma - (\mu - \epsilon)(\tau\gamma)^2\Big)\langle \nabla_2 f(x_k, y_k), \zeta_{2,k} \rangle - \frac{(\mu - \epsilon)(\tau\gamma)^2}{2}\|\zeta_{2,k}\|^2. \qquad (48)
\end{aligned}
$$

We next combine the bound on $f(x_{k+1}, y_{k+1}) - f(x_{k+1}, y_k)$ from (48) with the bound on $f(x_{k+1}, y_k) - f(x_k, y_k)$ from (44) to get a bound on $f(x_{k+1}, y_{k+1}) - f(x_k, y_k)$. Towards this end, let $\epsilon = \frac{1}{3}\mu$, then by combining (44) and (48),

and finally simplifying by grouping terms, we have the following:

$$
f(x_{k+1}, y_{k+1}) - f(x_k, y_k) \leq -\left(\gamma - \frac{3\gamma^2 L_1}{4}\right)\|\nabla_1 f(x_k, y_k)\|^2 - \gamma\langle\nabla_1 f(x_k, y_k), \zeta_{1,k}\rangle + \frac{3\gamma^2 L_1}{2}\|\zeta_{1,k}\|^2
$$

$$
+ \frac{3\gamma^2 L_2^2}{4\epsilon}\|\nabla_1 f(x_k, y_k)\|^2 + \frac{3\gamma^2 L_2^2}{2\epsilon}\|\zeta_{1,k}\|^2 + \left(\tau\gamma - \frac{(\mu - \epsilon)(\tau\gamma)^2}{2}\right)\|\nabla_2 f(x_k, y_k)\|^2
$$

$$
+ \left(\tau\gamma - (\mu - \epsilon)(\tau\gamma)^2\right)\langle\nabla_2 f(x_k, y_k), \zeta_{2,k}\rangle - \frac{(\mu - \epsilon)(\tau\gamma)^2}{2}\|\zeta_{2,k}\|^2
$$

$$
= -\left(\gamma - \frac{3\gamma^2 L_1}{4} - \frac{3\gamma^2 L_2^2}{4\epsilon}\right)\|\nabla_1 f(x_k, y_k)\|^2 - \gamma\langle\nabla_1 f(x_k, y_k), \zeta_{1,k}\rangle
$$

$$
+ \left(\frac{3\gamma^2 L_1}{2} + \frac{3\gamma^2 L_2^2}{2\epsilon}\right)\|\zeta_{1,k}\|^2 + \left(\tau\gamma - \frac{(\mu - \epsilon)(\tau\gamma)^2}{2}\right)\|\nabla_2 f(x_k, y_k)\|^2
$$

$$
+ \left(\tau\gamma - (\mu - \epsilon)(\tau\gamma)^2\right)\langle\nabla_2 f(x_k, y_k), \zeta_{2,k}\rangle - \frac{(\mu - \epsilon)(\tau\gamma)^2}{2}\|\zeta_{2,k}\|^2
$$

$$
\leq -\left(\gamma - \frac{3\gamma^2 L_1}{4} - \frac{9\gamma^2 L_2^2}{4\mu}\right)\|\nabla_1 f(x_k, y_k)\|^2 - \gamma\langle\nabla_1 f(x_k, y_k), \zeta_{1,k}\rangle
$$

$$
+ \left(\frac{3\gamma^2 L_1}{2} + \frac{9\gamma^2 L_2^2}{2\mu}\right)\|\zeta_{1,k}\|^2 + \left(\tau\gamma - \frac{\mu(\tau\gamma)^2}{3}\right)\|\nabla_2 f(x_k, y_k)\|^2
$$

$$
+ \left(\tau\gamma - \frac{2\mu(\tau\gamma)^2}{3}\right)\langle\nabla_2 f(x_k, y_k), \zeta_{2,k}\rangle. \tag{49}
$$

Observe that the inequality in (49) follows from the inequality

$$
-\frac{(\mu - \epsilon)(\tau\gamma)^2}{2}\|\zeta_{2,k}\|^2 = -\frac{\mu(\tau\gamma)^2}{3}\|\zeta_{2,k}\|^2 \leq 0.
$$

To proceed in bounding $f(x_{k+1}, y_{k+1}) - f(x_k, y_k)$, we derive upper bounds on coefficients on the right-hand side of (49). Define $\ell := L_1 + 3\kappa' L_2$ where $\kappa' := \frac{L_2}{\mu}$ and recall that $\gamma \leq \frac{1}{\ell}$ and $\tau\gamma \geq 4/\mu$. From the fact that $\gamma \leq 1/\ell$, we have that

$$
-\left(\gamma - \frac{3\gamma^2 L_1}{4} - \frac{9\gamma^2 L_2^2}{4\mu}\right) = -\gamma + \frac{3\gamma^2}{4}\left(L_1 + \frac{3L_2^2}{\mu}\right) = -\gamma + \frac{3\gamma^2\ell}{4} \leq -\gamma + \frac{3\gamma}{4} = -\frac{\gamma}{4}.
$$

This implies

$$
-\left(\gamma - \frac{3\gamma^2 L_1}{4} - \frac{9\gamma^2 L_2^2}{4\mu}\right)\|\nabla_1 f(x_k, y_k)\|^2 \leq -\frac{\gamma}{4}\|\nabla_1 f(x_k, y_k)\|^2. \tag{50}
$$

Moreover, from a direct simplification,

$$
\frac{3\gamma^2 L_1}{2} + \frac{9\gamma^2 L_2^2}{2\mu} = \frac{3\gamma^2}{2}\left(L_1 + \frac{3L_2^2}{\mu}\right) = \frac{3\gamma^2\ell}{2}.
$$

Hence,

$$
\left(\frac{3\gamma^2 L_1}{2} + \frac{9\gamma^2 L_2^2}{2\mu}\right)\|\zeta_{1,k}\|^2 = \frac{3\gamma^2\ell}{2}\|\zeta_{1,k}\|^2. \tag{51}
$$

Finally, using that $\tau\gamma \geq 4/\mu$, we also have

$$
\tau\gamma - \frac{\mu(\tau\gamma)^2}{3} = \tau\gamma - \frac{\mu\tau\gamma(\tau\gamma)}{3} \leq \tau\gamma - \frac{4\tau\gamma}{3} = -\frac{\tau\gamma}{3} \leq -\frac{\tau\gamma}{4}.
$$

Thus,

$$
\left(\tau\gamma - \frac{\mu(\tau\gamma)^2}{3}\right)\|\nabla_2 f(x_k, y_k)\|^2 \leq -\frac{\tau\gamma}{4}\|\nabla_2 f(x_k, y_k)\|^2. \tag{52}
$$

Combining (49) with (50), (51), and (52) gives the following one-step bound on the $\tau$-GDA update rule:

$$
f(x_{k+1}, y_{k+1}) - f(x_k, y_k) \leq -\frac{\gamma}{4}\|\nabla_1 f(x_k, y_k)\|^2 - \frac{\tau\gamma}{4}\|\nabla_2 f(x_k, y_k)\|^2
$$

$$
-\gamma\langle\nabla_1 f(x_k, y_k), \zeta_{1,k}\rangle + \left(\tau\gamma - \frac{2\mu(\tau\gamma)^2}{3}\right)\langle\nabla_2 f(x_k, y_k), \zeta_{2,k}\rangle + \frac{3\gamma^2\ell}{2}\|\zeta_{1,k}\|^2.
$$

Summing this inequality on both sides, we have that

$$f(x_k, y_k) - f(x_0, y_0) \leq -\frac{\gamma}{4}\Big(\sum_{t=0}^{k-1}\|\nabla_1 f(x_t, y_t)\|^2 + \tau\sum_{t=0}^{k-1}\|\nabla_2 f(x_t, y_t)\|^2\Big)$$
$$-\gamma\sum_{t=0}^{k-1}\langle\nabla_1 f(x_t, y_t), \zeta_{1,t}\rangle + \Big(\tau\gamma - \frac{2\mu(\tau\gamma)^2}{3}\Big)\sum_{t=0}^{k-1}\langle\nabla_2 f(x_t, y_t), \zeta_{2,t}\rangle + \frac{3\gamma^2\ell}{2}\sum_{t=0}^{k-1}\|\zeta_{1,t}\|^2.$$
$$(53)$$

Now, using the fact that $\zeta_{i,0}, \ldots, \zeta_{i,k-1}$ satisfy Condition 1 with norm-subGaussian parameter $\sigma_i$ for $i \in \mathcal{I}$, by Lemma 12 for each player $i \in \mathcal{I}$ and any $\iota > 0$ and $\lambda_i > 0$, there exists an absolute constant $c_i$ such that with probability at least $1 - e^{-\iota}$ we have:

$$\sum_{t=0}^{k-1}\langle\nabla_i f(x_t, y_t), \zeta_{i,t}\rangle \leq c_i\sigma_i^2\lambda_i\sum_{t=0}^{k-1}\|\nabla_i f(x_t, y_t)\|^2 + \frac{\iota}{\lambda_i}. \qquad (54)$$

Thus, as a result of (54), there exists an absolute constant $c_1$ such that with probability at least $1 - e^{-\iota}$ for any $\iota > 0$, we have

$$-\gamma\sum_{t=0}^{k-1} \leq \nabla_1 f(x_t, y_t), \zeta_{1,t}\rangle \leq \gamma\Big|\sum_{t=0}^{k-1}\langle\nabla_1 f(x_t, y_t), \zeta_{1,t}\rangle\Big|$$
$$\leq \gamma\Big(c_1\sigma_1^2\lambda_1\sum_{t=0}^{k-1}\|\nabla_1 f(x_t, y_t)\|^2 + \frac{\iota}{\lambda_1}\Big)$$
$$= \frac{\gamma}{8}\sum_{t=0}^{k-1}\|\nabla_1 f(x_t, y_t)\|^2 + 8c_1\gamma\sigma_1^2\iota \qquad (55)$$
$$= \frac{\gamma}{8}\sum_{t=0}^{k-1}\|\nabla_1 f(x_t, y_t)\|^2 + c_1'\gamma\sigma_1^2\iota. \qquad (56)$$

Note that in (55) $\lambda_1$ is being chosen as $\lambda_1 = 1/(8c_1\sigma_1^2)$ and in (56) the constant $c_1'$ is being defined as $c_1' := 8c_1$. Now, using that $\tau\gamma = \Gamma/\mu$ where $\Gamma > 4$ is an absolute constant, we have

$$\Big|\tau\gamma - \frac{2\mu(\tau\gamma)^2}{3}\Big| \leq \tau\gamma\Big|\frac{3 - 2\Gamma}{3}\Big|.$$

Combining this inequality with (54), there exists an absolute constant $c_2$ such that with probability at least $1 - e^{-\iota}$ for any $\iota > 0$, we have

$$\Big(\tau\gamma - \frac{2\mu(\tau\gamma)^2}{3}\Big)\sum_{t=0}^{k-1}\langle\nabla_2 f(x_t, y_t), \zeta_{2,t}\rangle \leq \Big|\tau\gamma - \frac{2\mu(\tau\gamma)^2}{3}\Big|\Big|\sum_{t=0}^{k-1}\langle\nabla_2 f(x_t, y_t), \zeta_{2,t}\rangle\Big|$$
$$\leq \tau\gamma\Big|\frac{3 - 2\Gamma}{3}\Big|\Big|\sum_{t=0}^{k-1}\langle\nabla_2 f(x_t, y_t), \zeta_{2,t}\rangle\Big|$$
$$\leq \tau\gamma\Big|\frac{3 - 2\Gamma}{3}\Big|\Big(c_2\sigma_2^2\lambda_2\sum_{t=0}^{k-1}\|\nabla_2 f(x_t, y_t)\|^2 + \frac{\iota}{\lambda_2}\Big)$$
$$= \frac{\tau\gamma}{8}\sum_{t=0}^{k-1}\|\nabla_2 f(x_t, y_t)\|^2 + 8c_2\Big|\frac{3 - 2\Gamma}{3}\Big|^2\tau\gamma\sigma_2^2\iota. \qquad (57)$$
$$= \frac{\tau\gamma}{8}\sum_{t=0}^{k-1}\|\nabla_2 f(x_t, y_t)\|^2 + c_2'\tau\gamma\sigma_2^2\iota. \qquad (58)$$

Note that in (57), $\lambda_2$ is being chosen as $\lambda_2 = 1/(8c_2\sigma_2^2|(3 - 2\Gamma)/3|)$, and in (58) the constant $c_2'$ is being defined as $c_2' := 8c_2|(3 - 2\Gamma)/3|^2$.

Now, using the fact that $\zeta_{1,0}, \ldots, \zeta_{1,k-1}$ satisfy Condition 1 with norm-subGaussian parameter $\sigma_1$, by Lemma 11 there exists an absolute constant $c_3$ such that with probability at least $1 - e^{-\iota}$ for any $\iota > 0$,

$$\sum_{t=0}^{k-1} \|\zeta_{1,t}\|^2 \le c_3 \sigma_1^2 (k + \iota). \tag{59}$$

Hence, by (59), there exists an absolute constant $c_3' := 3c_3/2$ such that with probability at least $1 - e^{-\iota}$ for any $\iota > 0$,

$$\frac{3\gamma^2 \ell}{2} \sum_{t=0}^{k-1} \|\zeta_{1,t}\|^2 \le c_3' \gamma^2 \ell \sigma_1^2 (k + \iota). \tag{60}$$

Finally, using (56), (58), and (60) in (53), we have that there exists an absolute constant $c := \max\{c_1', 2c_2', 2c_3'\}$ such that with probability at least $1 - 3e^{-\iota}$ for any $\iota > 0$,

$$f(x_k, y_k) - f(x_0, y_0) \le -\frac{\gamma}{4} \Big( \sum_{t=0}^{k-1} \|\nabla_1 f(x_t, y_t)\|^2 + \tau \sum_{t=0}^{k-1} \|\nabla_2 f(x_t, y_t)\|^2 \Big)$$

$$- \gamma \sum_{t=0}^{k-1} \langle \nabla_1 f(x_t, y_t), \zeta_{1,t} \rangle + \Big( \tau\gamma - \frac{2\mu(\tau\gamma)^2}{3} \Big) \sum_{t=0}^{k-1} \langle \nabla_2 f(x_t, y_t), \zeta_{2,t} \rangle + \frac{3\gamma^2 \ell}{2} \sum_{t=0}^{k-1} \|\zeta_{1,t}\|^2$$

$$\le -\frac{\gamma}{4} \Big( \sum_{t=0}^{k-1} \|\nabla_1 f(x_t, y_t)\|^2 + \tau \sum_{t=0}^{k-1} \|\nabla_2 f(x_t, y_t)\|^2 \Big)$$

$$+ \frac{\gamma}{8} \Big( \sum_{t=0}^{k-1} \|\nabla_1 f(x_t, y_t)\|^2 + \tau \sum_{t=0}^{k-1} \|\nabla_2 f(x_t, y_t)\|^2 \Big) + c_1' \gamma \sigma_1^2 \iota + c_2' \tau\gamma \sigma_2^2 \iota + c_3' \gamma^2 \ell \sigma_1^2 (k + \iota)$$

$$\le -\frac{\gamma}{8} \Big( \sum_{t=0}^{k-1} \|\nabla_1 f(x_t, y_t)\|^2 + \tau \sum_{t=0}^{k-1} \|\nabla_2 f(x_t, y_t)\|^2 \Big) + c\gamma \left( \sigma_1^2 (\gamma\ell k + \iota) + \tau\sigma_2^2 \iota \right).$$

Note the final inequality is obtained using that $c := \max\{c_1', 2c_2', 2c_3'\}$ and $\gamma \le 1/\ell$ to get

$$c_1' \gamma \sigma_1^2 \iota + c_3' \gamma^2 \ell \sigma_1^2 (k + \iota) = \max\{c_1', c_3'\} \gamma \sigma_1^2 (\iota + \gamma\ell k + \gamma\ell\iota) \le \max\{2c_1', 2c_3'\} \gamma \sigma_1^2 (\gamma\ell k + \iota),$$

so that

$$c_2' \tau\gamma \sigma_2^2 \iota + c_1' \gamma \sigma_1^2 \iota + c_3' \gamma^2 \ell \sigma_1^2 (k + \iota) \le c_2' \tau\gamma \sigma_2^2 \iota + \max\{2c_1', 2c_3'\} \gamma \sigma_1^2 (\gamma\ell k + \iota) \le c\gamma \left( \sigma_1^2 (\gamma\ell k + \iota) + \tau\sigma_2^2 \iota \right).$$

Finally, we remark that in the deterministic case that $\sigma_1^2 = \sigma_2^2 = 0$, the bound holds deterministically and simply reduces to

$$f(x_k, y_k) - f(x_0, y_0) \le -\frac{\gamma}{8} \Big( \sum_{t=0}^{k-1} \|\nabla_1 f(x_t, y_t)\|^2 + \tau \sum_{t=0}^{k-1} \|\nabla_2 f(x_t, y_t)\|^2 \Big).$$

This completes the proof. $\qquad\square$

## E.2 Proof of Corollary 2

In this appendix, we prove Corollary 2. We restate the result and then provide the proof.

**Corollary 2.** *Consider a nonconvex-SC zero-sum game $(f, -f)$ defined by $f \in C^2(\mathcal{Z}, \mathbb{R})$ satisfying Assumption 1 and Assumption 3. For any $\delta \in (0, 1)$, there exists $\gamma$ and $\tau$ satisying the conditions of Lemma 3 such that, with probability at least $1 - \delta$, starting from any initialization, at least half the iterates of (stochastic) $\tau$-GDA will be $\varepsilon$-critical points after $\widetilde{\mathcal{O}}(\varepsilon^{-2})$ and $\widetilde{\mathcal{O}}(\varepsilon^{-6})$ iterations in the deterministic and stochastic settings, respectively.*

*Proof of Corollary 2.* This proof primarily follows from Lemma 3. Specifically, Lemma 3 gives us a bound on the iterates of $\tau$-GDA decreasing the function value that will be used in a proof by contradiction. In particular, we assume that more than half the iterates after running the algorithm for $T$ steps are not $\varepsilon$-critical points. We then invoke Lemma 3 to conclude that if this was the case, then the function value would have decreased beyond the global minimum. Since this is not possible, it yields a contradiction, which then allows us to conclude that at least half of the iterates must be $\varepsilon$-critical points. We prove this result for the deterministic and stochastic settings independently following this general template.

**Deterministic Setting.** Suppose that $\gamma$ and $\tau$ are chosen as follows:

$$\gamma = 1/\ell \quad \text{and} \quad \tau = 4/(\mu\gamma).$$

Let the number of steps that the algorithm runs for be defined by

$$T := \frac{32(f(x_0, y_0) - f^*)}{\gamma\varepsilon^2} = \frac{32\ell(f(x_0, y_0) - f^*)}{\varepsilon^2} = \widetilde{\mathcal{O}}(\varepsilon^{-2}).$$

For the sake of contradiction, suppose that within $T$ steps, we have more than $T/2$ iterates for which $\max\{\|\nabla_1 f(x_k, y_k)\|, \|\nabla_2 f(x_k, y_k)\|\} \geq \varepsilon$. Then, by Lemma 3, taking $\sigma_1^2 = \sigma_2^2 = 0$, we have that:

$$f(x_T, y_T) - f(x_0, y_0) \leq -\frac{\gamma}{8} \sum_{k=0}^{T-1} \left( \|\nabla_1 f(x_k, y_k)\|^2 + \tau\|\nabla_2 f(x_k, y_k)\|^2 \right)$$

$$\leq -\frac{\gamma}{8} \sum_{k=0}^{T-1} \left( \|\nabla_1 f(x_k, y_k)\|^2 + \|\nabla_2 f(x_k, y_k)\|^2 \right) \tag{61}$$

$$\leq -\frac{\gamma}{8} \sum_{k=0}^{T-1} \max\{\|\nabla_1 f(x_k, y_k)\|^2, \|\nabla_2 f(x_k, y_k)\|^2\} \tag{62}$$

$$\leq -\frac{\gamma T \varepsilon^2}{16}. \tag{63}$$

Observe that in (61) we have applied the fact that $\tau \geq 1$, in (62) we have used that the sum of positive numbers is greater than the max of the numbers, and in (63) we have invoked the assumption that at least $T/2$ iterates satisfy $\max\{\|\nabla_1 f(x_k, y_k)\|, \|\nabla_2 f(x_k, y_k)\|\} \geq \varepsilon$.

We now show that this implies a contradiction. In particular, the bound in (63) implies that

$$f(x_T, y_T) \leq f(x_0, y_0) - \frac{\gamma T \varepsilon^2}{16} = f(x_0, y_0) - \frac{\gamma\varepsilon^2}{16} \frac{32(f(x_0, y_0) - f^*)}{\gamma\varepsilon^2} < f(x_0, y_0) - f(x_0, y_0) - f^* = f^*.$$

This is not possible and yields a contradiction to the assumption that we have more than $T/2$ iterates for which $\max\{\|\nabla_1 f(x_k, y_k)\|, \|\nabla_2 f(x_k, y_k)\|\} \geq \varepsilon$. Hence, at most $T/2$ iterates have $\max\{\|\nabla_1 f(x_k, y_k)\|, \|\nabla_2 f(x_k, y_k)\|\} \geq \varepsilon$ and thus at least $T/2$ iterates must be $\varepsilon$-critical points if the algorithm runs for $T = \widetilde{\mathcal{O}}(\varepsilon^{-2})$ iterations.

**Stochastic Setting.** Suppose that $\gamma$ and $\tau$ are chosen as follows:

$$\gamma := \frac{1}{\iota^2 \ell \mathcal{R}} \quad \text{and} \quad \tau = \frac{4}{\mu\gamma} \quad \text{where} \quad \mathcal{R} := 1 + \frac{\max\{\sigma_1^2, \sigma_2^2\}}{\varepsilon^2}.$$

Moreover, $\iota > 0$ is an absolute constant to be chosen later. Let the number of steps that the algorithm runs for be defined by

$$T := \frac{\tau(f(x_0, y_0) - f^*)}{\ell\gamma\varepsilon^2} = \frac{(f(x_0, y_0) - f^*)\ell\iota^4\mathcal{R}^2}{\mu\varepsilon^2} = \widetilde{\mathcal{O}}(\varepsilon^{-6}). \tag{64}$$

For the sake of contradiction, suppose that within $T$ steps, we have more than $T/2$ iterates for which $\max\{\|\nabla_1 f(x_k, y_k)\|, \|\nabla_2 f(x_k, y_k)\|\} \geq \varepsilon$. Then, by Lemma 3, there exists an absolute constant $c$ such that with probability at least $1 - 3e^{-\iota}$ for any $\iota > 0$, we have that:

$$f(x_T, y_T) - f(x_0, y_0) \leq -\frac{\gamma}{8} \sum_{k=0}^{T-1} \left( \|\nabla_1 f(x_k, y_k)\|^2 + \tau\|\nabla_2 f(x_k, y_k)\|^2 \right) + c\left( \gamma\sigma_1^2(\gamma\ell T + \iota) + \tau\gamma\sigma_2^2\iota \right)$$

$$\leq -\frac{\gamma}{8} \sum_{k=0}^{T-1} \left( \|\nabla_1 f(x_k, y_k)\|^2 + \|\nabla_2 f(x_k, y_k)\|^2 \right) + c\left( \gamma\sigma_1^2(\gamma\ell T + \iota) + \tau\gamma\sigma_2^2\iota \right) \tag{65}$$

$$\leq -\frac{\gamma}{8} \sum_{k=0}^{T-1} \max\{\|\nabla_1 f(x_k, y_k)\|^2, \|\nabla_2 f(x_k, y_k)\|^2\} + c\left( \gamma\sigma_1^2(\gamma\ell T + \iota) + \tau\gamma\sigma_2^2\iota \right) \tag{66}$$

$$\leq -\frac{\gamma T \varepsilon^2}{16} + c\gamma\left( \sigma_1^2(\gamma\ell T + \iota) + \tau\sigma_2^2\iota \right). \tag{67}$$

Observe that in (65) we have applied the fact that $\tau \geq 1$, in (66) we have used that the sum of positive numbers is greater than the max of the numbers, and in (67) we have invoked the assumption that at least $T/2$ iterates satisfy $\max\{\|\nabla_1 f(x_k, y_k)\|, \|\nabla_2 f(x_k, y_k)\|\} \geq \varepsilon$.

Define $\sigma^2 := \max\{\sigma_1^2, \sigma_2^2\}$. We now continue upper bounding (67) the latter term on the right-hand side. To begin, using that $\sigma^2 := \max\{\sigma_1^2, \sigma_2^2\}$ and $\tau \geq 1$, we have

$$c\gamma \left(\sigma_1^2(\gamma\ell T + \iota) + \tau\sigma_2^2\iota\right) \leq c\gamma\sigma^2(\gamma\ell T + \iota + \tau\iota) \leq c\gamma\sigma^2\tau\iota\left(\frac{\gamma\ell T}{\tau\iota} + 2\right).$$

Observe that since $\mathcal{R} = 1 + \sigma^2/\varepsilon^2 \geq \sigma^2/\varepsilon^2$, we have

$$\gamma\tau\sigma^2\iota = \frac{\tau\sigma^2\iota}{\ell\iota^2\mathcal{R}} = \frac{\tau\sigma^2}{\iota\ell\mathcal{R}} \leq \frac{\tau\varepsilon^2}{\ell\iota}.$$

Hence, combining the previous inequalities, we get

$$c\gamma \left(\sigma_1^2(\gamma\ell T + \iota) + \tau\sigma_2^2\iota\right) \leq \frac{c\tau\varepsilon^2}{\ell\iota}\left(\frac{\gamma\ell T}{\tau\iota} + 2\right) = \frac{c\gamma T\varepsilon^2}{\iota^2} + \frac{2c\tau\varepsilon^2}{\ell\iota} = \frac{c\gamma T\varepsilon^2}{\iota^2} + \frac{2c\tau\gamma\varepsilon^2}{\ell\gamma\iota}.$$

Now, since

$$\frac{\tau}{\gamma\ell} \leq \frac{\tau(f(x_0, y_0) - f^*)}{\gamma\ell\varepsilon^2} = T,$$

by choosing $\iota$ as a sufficiently large absolute constant, we get

$$c\gamma \left(\sigma_1^2(\gamma\ell T + \iota) + \tau\sigma_2^2\iota\right) \leq \frac{c\gamma T\varepsilon^2}{\iota^2} + \frac{2c\tau\gamma\varepsilon^2}{\ell\gamma\iota} \leq \frac{c\gamma T\varepsilon^2}{\iota^2} + \frac{2c\gamma T\varepsilon^2}{\iota} \leq \frac{\gamma T\varepsilon^2}{32}.$$

Thus, returning to (67), we have that

$$f(x_T, y_T) - f(x_0, y_0) \leq -\frac{\gamma T\varepsilon^2}{16} + c\gamma \left(\sigma_1^2(\gamma\ell T + \iota) + \tau\sigma_2^2\iota\right) \leq -\frac{\gamma T\varepsilon^2}{16} + \frac{\gamma T\varepsilon^2}{32} = -\frac{\gamma T\varepsilon^2}{32}.$$

We now show that this implies a contradiction. In particular, this bound implies that

$$\begin{aligned}
f(x_T, y_T) &\leq f(x_0, y_0) - \frac{\gamma T\varepsilon^2}{32} \\
&\leq f(x_0, y_0) - \frac{\gamma\varepsilon^2}{32}\frac{\tau(f(x_0, y_0) - f^*)}{\ell\gamma\varepsilon^2} \\
&= f(x_0, y_0) - \frac{4\iota^2\ell\mathcal{R}}{32\mu}(f(x_0, y_0) - f^*) \\
&< f(x_0, y_0) - f(x_0, y_0) - f^* \\
&= f^*.
\end{aligned}$$

This is not possible and yields a contradiction to the assumption that we have more than $T/2$ iterates for which $\max\{\|\nabla_1 f(x_k, y_k)\|, \|\nabla_2 f(x_k, y_k)\|\} \geq \varepsilon$. Hence, with probability at least $1 - 3e^{-\iota}$, at most $T/2$ iterates have $\max\{\|\nabla_1 f(x_k, y_k)\|, \|\nabla_2 f(x_k, y_k)\|\} \geq \varepsilon$. Thus, with probability at least $1 - 3e^{-\iota}$, at least $T/2$ iterates must be $\varepsilon$-critical points if the algorithm runs for $T = \widetilde{\mathcal{O}}(\varepsilon^{-6})$ iterations. This statement then holds with probability at least $1 - \delta$ for the given $\delta \in (0, 1)$ by selecting a sufficient large constant $\iota > 0$. This completes the proof. $\square$