# OpenReview forum: "Global Convergence  to Local Minmax Equilibrium in Classes of Nonconvex Zero-Sum Games"
_NeurIPS.cc/2021/Conference — NeurIPS 2021 Poster_

### Official Review · Reviewer_ZYYQ · 2021-07-16

**Rating:** 6
**Confidence:** 5

**Summary:**

This paper studies gradient descent ascent learning dynamics with timescale separation in unconstrained continuous action zero-sum games. The minimizing player faces a nonconvex optimization problem, and the maximizing player optimizes a Polyak-Łojasiewicz (PŁ) or strongly-concave (SC) objective. Like [15], this paper assesses convergence to Stackelberg equilibria instead of only notions of stationary. A differential Stackelberg equilibrium corresponds to a joint strategy at which the minimizing player is at a local optimum to its choice variable along the best response curve of the maximizing player, and the maximizing player is at a local optimum concerning its choice variable. In pursuit of this goal, the authors prove that the only locally stable points of the continuous-time limiting system correspond to strict local Stackelberg equilibria in nonconvex-PŁ and nonconvex-SC games. Specifically, this paper exploits timescale separation to construct a potential function of the nonconvex-PŁ game that gives a global asymptotic almost-sure convergence guarantee to the set of strict local Stackelberg equilibria with the stability and asymptotic saddle avoidance. Moreover, this paper shows that nonconvex-SC games do gradient descent ascent provably converge in an almost sure asymptotic sense and a finite time escape time from saddle points.

Overall, this paper gives some theoretical contributions, with the caveat for the clarifications of the assumptions and the shortage of empirical studies.

**Limitations And Societal Impact:**

Yes

**Main Review:**

For the theory, there are a few assumptions that need clarification and further clarification on novelty. Theorem 1 is established for non-convex PŁ games. Theorem 1 looks like a corollary of  [Theorem 1, 15].  Moreover, Theorem 1 does not give the choice of $\tau_*$. Although the PŁ and SC conditions bring the global asymptotic convergence of $\tau$-GDA and $\tau$-PGDA, the results of Sec. 5 and 6 are familiar with non-convex optimization researchers. Therefore, the authors should include the results of [15]  in Table 1.

Moreover, this paper ignores the empirical studies at all. The authors claim that fair classification, adversarial training, and distributionally robust optimization belong to the class of games they consider. None of these applications are conducted in the paper. It intensified the reviewers’ concerns about the above assumptions. Although the code is in the supplementary material, numerical visualization is an indispensable part.



**Time Spent Reviewing:**

12

---

> ### Author Response · Authors · 2021-08-06
> **Response to Reviewer ZYYQ**
>
> Thank you for your time spent reviewing our paper. We appreciate your comments and hopefully we can clear up your concerns.
>
> Theorem 1 is not a corollary of Theorem 1 from [15].  The key contribution of our Theorem 1 is part 1, which says that in nonconvex-PL/SC zero-sum games, any critical point that is not a strict local minmax is not stable for all $\tau >0$. In other words, critical points that are not strict local minmax are never stable no matter the choice of timescale separation. We crucially exploit the quadratic growth condition of PL/SC functions in the proof to obtain part 1 of our theorem, whereas the result from Theorem 1 from [15] is about stability and not instability and is focused on the more general setting of nonconvex-nonconcave zero-sum games with weaker characterizations. Thus, our result is a novel and actually requires different proof methods. Moreover, our Theorem 1 is also key to the rest of our results that build off of it since it implies that the only critical points that could possibly be stable are strict local minmax.
>
> Theorem 1 from [15] says that there exists a $\tau^{\ast}\in (0, \infty)$ such that $z=(x, y)$ is stable for all $\tau \in [\tau^{\ast}, \infty)$ if and only if $z$ is a strict local minmax. Part 2 of our Theorem 1 strengthens the result from Theorem 1 of [15] to say that in nonconvex-PL/SC zero-sum games, that after the minimum $\tau^{\ast}$ such that $z$ is stable, $z$ is stable for all $\tau \geq \tau^{\ast}$. In particular, it strengthens the result by establishing that a point $z$ cannot become stable, then go to being unstable, and then return to being stable when it is a strict local minmax equilibrium. The fact that such a $\tau^{\ast}$ exists such that for all $\tau >\tau^{\ast}$, a strict local minmax is stable can be determined by Theorem 1, from [15]. Primarily, the point of stating Part 2 of Theorem 1 is to make clear that there is a finite timescale separation such that any strict local minmax can be stable, but the most novel and important contribution of our Theorem 1 is Part 1.
>
> We believe this is made fairly clear in the paragraph after our Theorem 1, but let us know if you have any remaining questions or concerns in this regard.
>
> Theorem 1 does not include a choice of $\tau^{\ast}$ for several reasons. First, the value of $\tau^{\ast}$ is dependent on a particular equilibrium. That is, it is not necessarily uniform across all the strict local minmax equilibrium. Moreover, for a particular equilibrium $z=(x, y)$, the value of $\tau^{\ast}$ can be obtained via the construction in Theorem 1 of [15]. Notably, this construction depends on properties of the equilibrium.
>
> This being said, the potential functions that are constructed imply a choice of $\tau$ such that a strict local minmax equilibrium is ensured to be stable. Specifically, as stated in Lemma 1 and Theorem 2, for $\tau>7\kappa^2$, there is a potential function. As can be seen in the proofs of Lemma 1 and Theorem 2 in Appendix D.1 and D.2, the potential function only stops decreasing at critical points (that is $z=(x,y)$ such that $\nabla_xf(x, y)=0$, $\nabla_yf(x, y)=0$). Since we have shown in Lemma 2 that the discrete-time update converges to strict saddles on a set of measure zero and in Theorem 1 that points that are not strict local minmax are saddles for any choice of timescale $\tau$, this together implies that with this choice of $\tau$ that the update will almost surely converge to a strict local minmax. Thanks for pointing this out and we will make it more clear in the paper.
>
> We would like to respectively push back against your claim that ``Although the PL and SC conditions bring the global asymptotic convergence of $\tau$-GDA and$\tau$-PGDA, the results of section 5 and 6 are familiar with nonconvex optimization researchers’’. There are no such results of global convergence to only strict local minmax equilibrium for nonconvex-PL/SC zero-sum games. Moreover, there has actually been no finite-time saddle escaping result for zero-sum games. We ask that you please carefully consider this since it seems to be discounting the results without any basis.
>
> We believe that this work stands on its own without empirical results, given that it is already nearly 50 pages with a significant amount of theoretical analysis. Moreover, we study a common algorithm which is already used in practice, so it is not entirely clear how empirical results would strengthen the paper. We note that a significant proportion of related papers on nonconvex-PL/SC zero-sum games do not contain empirical studies and also that often when they do the actual experiments reflect settings where the theoretical assumptions are violated.
>
> The practical examples were intended to be illustrative and it was mentioned following them (see lines 78-79) that the formulations can be converted to analogous unconstrained problems with appropriate modifications. We would be happy to make it more concrete how this would be done if you believe it would make the presentation of the paper stronger in your view. For instance, consider example 1. The problem formulation stated in the paper was
>      $$\min_{W} \max_{i=1,\ldots, m} L_i(W)$$
> where we can treat $L_i(W)$ as the cross entropy loss for class $i\in [m]$ and recall that $W$ is denoting the weights of the neural network. This problem can be similarly formulated as an unconstrained nonconvex-strongly-concave problem by using the formulation
> $$\min_{W} \max_{\lambda \in  R^m}\sum_{i=1}^m \lambda_i L_i(W) - \nu ||\lambda||_2^2$$
> where $\nu>0$ is a regularization parameter. The inner maximization is strongly concave since it is a sum of a linear function and a strongly concave function. Thus, the overall problem is a nonconvex-strongly-concave zero-sum game. Note that since the losses are always positive, clearly this objective still encourages the maximizer to put most weight on the the class where the loss is highest. Moreover, strictly convex regularizers (such as $||\lambda||_2^p$ for $p\geq 2$ even) could be added to make the penalty sharper or smooth strongly-convex (concave) approximations of the $\ell_1$ norm could be used as a regularizer to enforce the sparsity.
> There are many other adjustments that can be made, but we gave this example to give the simplest formulation which would result in an analogous objective. Similar reformulations could be made for examples 2 and 3. We note that relaxations of this type from constrained problems to unconstrained problems are also common in normal function optimization.
>
>
> Thanks for your consideration of our response! Please let us know if you have any remaining concerns or if there is anything that would help you be able to recommend this paper as an accept.
>
> [15] Fiez, Ratliff, ICLR 2021. Local Convergence Analysis of Gradient Descent-Ascent with Finite Timescale Separation.

---

> > ### Author Response · Authors · 2021-08-27
> > **Reviewer ZYYQ: did our response clarify your questions?**
> >
> > Dear reviewer ZYYQ, thank you for your effort reviewing our paper. We wanted to check in and see if the points that you mentioned needed clarification have been resolved by our response?

---

### Official Review · Reviewer_kefj · 2021-07-18

**Rating:** 6
**Confidence:** 4

**Summary:**

The authors focus on the analysis of gradient descent ascent dynamics with timescale separation to solve unconstrained min-max optimization for landscapes that are non-convex on the min player (say x) and strongly concave or satisfy the PL condition for the max player. They focus on the stability analysis and convergence rates of GDA (and its perturbed variant) to the local minimax points (defined in Jin et al paper "What is locally..."). The definition of local min-max points is that the max player plays local best response and the min player chooses a local minimum for the function \max_{y \in B} f(.,y) where B is a ball of small radius. The authors show that local min-max points are stable/ attracting for GDA dynamics; the learning rate of the max player must be naturally larger than that of the min player in GDA (in some sense y updates more often). They also show convergence in \tilde{O}(1/\eps^2) of GDA to an \esp- local min-max in the case of non-convex stronly concave landscapes and non-convex, PL and \tilde{O}(1/\eps^4) of SGDA for the case of non-convex strongly concave landscapes.

**Limitations And Societal Impact:**

The paper is theoretical and the reviewer does not see any societal impact.

**Main Review:**

I will start by saying that the fact that local minimax points are stable for non-convex strong concave or PL landscapes is not suprising since effectively the hessian with respect to y is negative definite (invertible) and all degenerecies are avoided (we can decide about stability by checking the eigenvalues of the hessian and not cheking third order terms) and moreover use the multiplicative property of the determinant and Schur complement to derive a relation between the eigenvalues of the Hessian, the lower right block matrix of the Hessian and its schur complement. What seems a bit more interesting is that all the other critical points are unstable (I feel that every critical point that is unstable, it is unstable because of the x variables (for the examined landscapes), is this true? The fact that we have convergence to local minimax is not surprising (similar result appears in https://arxiv.org/abs/1902.08297), I like though that the authors use potential argument to show this fact.

The result about SGDA follows standard machinery for perturbed GD dynamics (see Jin et al 2017) and the \tilde{O}(1/\eps^4) was expected. Overall, I think the paper is an interesting but not super strong. Can also the authors argue about constructing functions (non-convex in x and PL condition in y) with a critical point the corresponding matrix of which has at least one imaginary eigenvalue (for the continuous time dynamics)? Which parts of the proofs carry over for constrained min-max optimization?

**Time Spent Reviewing:**

5

---

> ### Author Response · Authors · 2021-08-06
> **Response to Reviewer kefj**
>
> Thank you for your time spent reviewing and the helpful feedback you have provided.
>
> There is a major point we would like to clear up first that is critical. In your review, you say that “The fact that we have convergence to local minimax is not surprising (similar result appears in https://arxiv.org/abs/1902.08297), ”. These results you refer to only give convergence to what the authors refer to as `first-order nash equilibrium’, which in the unconstrained setting is just a critical point (meaning $z=(x,y)$ such that $\nabla_xf(x, y)=0$, $\nabla_yf(x, y)=0$) which corresponds to any stationary point of the dynamics. That is, they are only showing convergence to critical points of the dynamics (also with a different algorithm) and they are saying nothing about whether the critical point is any type of equilibrium (specifically a local minmax point), which requires assessing the critical points in terms of second order conditions. So just to reiterate, their results give convergence guarantees to critical points (meaning they only satisfy first order conditions), which could even be local-maxmax points, whereas we are giving a much stronger convergence guarantee to only strict local minmax. This is actually our primary motivation for the work in this paper, so it is a crucial distinction and in fact the key significance of our results we hope is now clear.
>
> Given our interpretation of your comment about the local minmax being stable not being surprising, it appears you are claiming that since $\text{det}(J_{\tau}(z)) = \text{det}(S_1(J(z)))\text{det}(-\tau D_2^2 f(z))$, it is straightforward that $z=(x, y)$ is stable when $S_1(J(z))$ and $-D_2^2f(z)$ are both positive definite in nonconvex-PL/SC games. This is in fact not true, since stability cannot be determined by the sign of the determinant of the Jacobian. The reason is that $\text{det}(J_\tau(z))$ being positive does not mean all the eigenvalues have positive real parts, which is the requirement for stability of z with respect to the dynamics (note that stability normally is defined in terms of negative real parts of the local linearization, here this is equivalent to having positive real parts of the Jacobian $J_\tau$. This is because the determinant is the product of the eigenvalues, so there could be negative eigenvalues of $J_\tau$ that multiply together so that the product of all eigenvalues is positive. It has also already been established that for a fixed choice of $\tau$, there can exist strict local minmax equilibrium that are not stable (see e.g., [15, 21] ). For a simple example, consider $f(x, y)= -1.5*x^2 + 4xy- y^2$. The only critical point is $z=(0, 0)$. The Jacobian with $\tau=1$ is $J_{1}(z)=[-3, 4; -4, 2]$ which has eigenvalues of $-0.5\pm i\sqrt{39}$ so it is not stable. But it is a strict local minmax since $-D_2^2f(z)=2>0$ and $S_1(J(z))=-3+(4^2)/2=5>0$. However, for all $\tau>1.5$, $z$ is in fact stable.
>
> Thank you for your appreciation of the result that all points that are not local minmax are not stable for any choice of $\tau$. We agree this is interesting since it indicates that no matter the choice of timescale separation, the only points that can possibly be stable are strict local minmax equilibrium. This is not only because of the $x$ players variables, but it actually has to do with interaction structure between the players, and the cost structure with respect to only the $y$ players variables. Essentially the result is saying that given $-D_2^2f(z)$ is positive definite and $S_1(J(z))=D_1^2f(z)-D_{12}f(z)[D_2^2f(z)]^{-1}D_{12}^{\top}f(z)$ is not positive definite, then $-J_{\tau}(z)$ cannot be a Hurwitz matrix. You can see from the given information that the Schur complement not being positive definite can imply things about each matrix in the game Jacobian, so it is not solely dependent on the $x$ player variables. That being said, given that $-D_2^2f(z)$ is positive definite, when $S_1(J(z))$ is not positive definite, we at least know that $D_1^2f(z)$ is not positive definite. But again, it is important to recall that $D_1^2f(z)$ can be not positive definite, and $z$ can still be a strict local minmax since $S_1(J(z))$ may still be positive definite. In fact, the example above in the previous paragraph is exactly like this.
>
>
> We also want to note that our result about PGDA uses techniques from the perturbed GD dynamics, there are actually some major distinctions so it is not necessarily an expected result. In particular, a key step is actually showing that the function itself can be a potential function for gradient descent-ascent, whereas this has not been a typical potential function in other analysis's of nonconvex-strongly-concave zero-sum games. Then there is some additional challenges that result from the nonsymmetric nature of the Jacobian of the dynamics, so this is a pretty significant departure from just applying the result for perturbed gradient descent. More discussion of this was provided in both Section 6 and the beginning of Appendix E in the paragraph titled `` Differences from the single player optimization setting’’ at line 1062.  Moreover, to obtain this result necessarily needs the stability and instability characterization of Theorem 1. Thus, we think that you may be discounting this result some and we would ask you to consider that.
>
> In your review, you say that `overall, I think the paper is interesting but not super strong’. We hope that the comments above make you feel differently. If they did not, could you explain some more to us why you feel this way? It was not clear from your review where this statement is coming about from.
>
> There are certainly games where there is eigenvalues with imaginary parts in the Jacobian, in fact this is the norm and not an exception. As one simple convex-concave game, consider the zero sum game  $(f,-f)$ defined by $f(x, y)= -0.1*x^2 + 2xy- y^2$.  This is non-convex in $x$ and strongly-concave in $y$ (hence, PL). Then, with $\tau=1$, $J_{\tau}(0, 0)=[-0.2, 2; -2, 2]$ and the eigenvalues are $0.9\pm- 1.67i$. We note this is not a degenerate example; it is straightforward to construct a parametric version of this game such that there is a continuum of games with imaginary eigenvalues.  We are not sure about the significance of this inquiry, but are happy to provide more examples if you could please elaborate further.
>
> It is difficult to say in short order what parts of the proofs would carry over to the constrained setting, so we would rather keep the discussion focused on the unconstrained setting which is the setting we are studying in this paper. That being said, we believe the constrained setting is a really interesting direction of work that will certainty be pursued in the future by us or others and likely this work will be a key step toward developing those results.
>
> Thanks for your consideration of our response! Please let us know if you have any remaining concerns or if there is anything that would help you be able to recommend this paper as an accept.
>
> [15] Fiez, Ratliff, ICLR 2021. Local Convergence Analysis of Gradient Descent-Ascent with Finite Timescale Separation.
>
> [21] Jin et al. ICML 2020. What is Local Optimality in Nonconvex-Nonconcave Minimax Optimization.

---

> > ### Comment · Reviewer_kefj · 2021-08-25
> > **reviewer's response**
> >
> > The authors in https://arxiv.org/abs/1902.08297 define the first order Nash equilibrium in such a way that for the unconstrained case, it boils down to critical points, so I agree with the authors. But this is essentially half of your result, i.e., establishing convergence to critical points first. Nevertheless, convergence to critical points was already established. In this paper, the new thing is the stability analysis so that you can argue that strict saddles are not Stackelberg equilibrium. So as far as I am concerned, the new contribution is establishing that the strict saddles are not strict local minmax. As far as the stability analysis, all I was saying is that e.g. (see the paper "On Solving Minimax Optimization Locally: A Follow-the-Ridge Approach", page 4 that stability analysis (of the Hessian in which you have minus sign in the first row block) boils down to the stability analysis of the max player Hessian part (say the function is non-convex concave) times the Schur complement and this is true as you can write it as a product (in case of characteristic polynomial this uses also that det of the product is product of det). Now positive definiteness of the Hessian of the max player and the Schur complement is what you need (again matrix M in page 4 of On Solving Minimax Optimization Locally: A Follow-the-Ridge Approach is very illustrating.
> >
> > I will keep my score because I feel that the contribution is ok but I am not super enthusiastic about this paper.

---

> > > ### Author Response · Authors · 2021-08-25
> > > **2nd Response to KEFJ: Clarifying Novelty and Analysis Methods**
> > >
> > > We appreciate you taking the time to engage and respond in this review process further.
> > >
> > > Just to begin, it seems that your lack of enthusiasm stems from a perception that some of the results follow directly from past work. This is not the case, so we will try to clarify now since from our perspective it seems there is some misinterpretations of the relationships between some past results and those which we present in our paper.
> > >
> > > We would like to point out that the paper you are referring to (https://arxiv.org/abs/1902.08297) shows convergence to critical points for a different algorithm. Namely, alternating gradient descent-ascent with k steps of unrolling the maximization procedure and the result requires that k is sufficiently large to get convergence to critical points. Even if k could be taken to be k=1 (which it can’t for the result in that paper) it would still be a different algorithm than gradient descent-ascent since it would have alternating updates. Our result is showing a potential function to show convergence to critical points using gradient descent-ascent (that is, each update step is taken simultaneously). This requires different proof techniques, and is a new result since it is for a different algorithm. So, this step that helps lead to the global convergence guarantee to local minmax was not already done.
> > >
> > > It appears that you may be conflating the J matrix on page 4 of the follow the ridge paper with the J used in our paper. The J matrix in the follow the ridge paper is the Jacobian of the dynamics they study in Algorithm 1 (FTR), which are not the gradient descent-ascent dynamics, but instead includes an extra term in the update for the maximizing player which leads to a different structure for the local linearization (Jacobian). This can be seen on page 4 of the follow the ridge paper where the J matrix is defined as (in our notation) by:
> > >
> > > $$J_{\text{FTR}}(z) = I - \gamma_1 \begin{bmatrix} I & 0 \\\ -[D_2^2f(z)]^{-1}D_{21}f(z) & I\end{bmatrix} \begin{bmatrix} D_1^2f(z) & D_{12}f(z) \\\ -\tau D_{21} f(z) & -\tau D_{2}^2f(z)\end{bmatrix}$$
> > >
> > > In comparison, the Jacobian for the GDA dynamics, is defined as
> > >
> > > $$J_{\text{GDA}}(z) = I - \gamma_1 \begin{bmatrix} D_1^2f(z) & D_{12}f(z) \\\ -\tau D_{21} f(z) & -\tau D_{2}^2f(z)\end{bmatrix}$$
> > >
> > > The FTR dynamics and stability analysis crucially rely on the transformation of coordinates which you can see as the preconditioner of the matrix which would be our Jacobian (i.e. the Jacobian of the continuous time gradient descent-ascent)
> > >
> > > As a result, the eigenvalues of the Jacobian of the FTR dynamics are exactly equal to the union of the eigenvalues of the schur complement of our J, which is given by $D_1^2f(z)-D_{12}f[D_2^2f(z)]^{-1}D_{21}f(z)$ and $-\tau D_2^2f(z)$. This is not the case for the Jacobian of the gradient descent-ascent dynamics. This is why it is not direct as you are claiming.
> > >
> > > Also, it seems that you are not acknowledging the finite-time escaping of saddles at all that we have proved, which is the first result of its kind for gradient-based learning in games.
> > >
> > > Thanks for responding, and let us know if helps clarify the novelty of the contributions to you.

---

### Official Review · Reviewer_gXSR · 2021-07-19

**Rating:** 7
**Confidence:** 4

**Summary:**

In this paper, the authors study the problem of solving specific classes of non-convex min-max optimization problems. More specifically, they consider the problem when the maximizing player optimizes a Polyak-Lojasiewicz or strongly concave objective. In contrast to existing work, the paper focuses on characterizing the convergence of gradient-based algorithms to local Nash Equilibria that are meaningful in game theoretical settings. The authors show that for non-convex PL games and under a strict saddle property, $\tau$-GDA  asymptotically converges to strict local minmax equilibria almost surely. For the class of non-convex strongly concave games, the authors show that $\tau$-PGDA converges to an approximate $epsilon$-local minmax equilibria. For both cases the complexity rate of the algorithms are provided.

**Limitations And Societal Impact:**

Yes

**Main Review:**

The paper studies a significant topic that belongs to a vibrant research field. The paper is easy to follow and clearly written.
Despite being extensively studied in the past few years, the authors distinguish the paper by obtaining more stringent convergence guarantees. More specifically, they obtain convergence guarantees in regards to differential Stackelberg equilibria compared to assessing convergence to approximate stationary points in past works. The results presented are not very surprising since Danskin's (or a Danskin's like) theorem holds for these classes of non-convex settings. However, the results are not trivial and cleanly developed.

I have the following concerns:

1. Some practical examples presented in section 1.2 do not apply to the non-convex PL/SC settings. Also in equation (3), should it be f_i?

2. Line 162: I believe the gradient does not always exist.

3. The authors define the constrained min-max problem, but then study the unconstrained problem. Also why f^* is defined in line 379.

4. A major concern is that the paper lacks experimental results. Motivating the class of problem under study with practical examples and  illustrating the results through empirical results would help improve the paper.




**Time Spent Reviewing:**

6

---

> ### Author Response · Authors · 2021-08-05
> **Response to Reviewer gXSR**
>
> Thank you for your time spent reviewing and the helpful feedback you have provided.
>
> Regarding your questions/comments in order:
>
> 1a. The practical examples were intended to be illustrative and it was mentioned following them (see lines 78-79) that the formulations can be converted to analogous unconstrained problems with appropriate modifications. We would be happy to make it more concrete how this would be done if you believe it would make the presentation of the paper stronger in your view. For instance, consider example 1. The problem formulation stated in the paper was
>      $$\min_{W} \max_{i=1,\ldots, m} L_i(W)$$
> where we can treat $L_i(W)$ as the cross entropy loss for class $i\in [m]$ and recall that $W$ is denoting the weights of the neural network. This problem can be similarly formulated as an unconstrained nonconvex-strongly-concave problem by using the formulation
> $$\min_{W} \max_{\lambda \in  R^m}\sum_{i=1}^m \lambda_i L_i(W) - \nu ||\lambda||_2^2$$
> where $\nu>0$ is a regularization parameter. The inner maximization is strongly concave since it is a sum of a linear function and a strongly concave function. Thus, the overall problem is a nonconvex-strongly-concave zero-sum game. Note that since the losses are always positive, clearly this objective still encourages the maximizer to put most weight on the the class where the loss is highest. Moreover, strictly convex regularizers (such as $||\lambda||_2^p$ for $p\geq 2$ even) could be added to make the penalty sharper or smooth strongly-convex (concave) approximations of the $\ell_1$ norm could be used as a regularizer to enforce the sparsity.
> There are many other adjustments that can be made, but we gave this example to give the simplest formulation which would result in an analogous objective. Similar reformulations could be made for examples 2 and 3. We note that relaxations of this type from constrained problems to unconstrained problems are also common in normal function optimization.
>
> 1b. Yes, in equation (3) it should be f_i.
>
> 2. This gradient does always exist in the nonconvex-PL/SC problems since Danskin’s (or a Danskin-like) Theorem does hold (for the PL-case, see Lemma A.5 in https://arxiv.org/pdf/1902.08297.pdf and note that this would be evaluating the derivative at any maximizer among the set of maximizers). We are happy to edit the paper and make this more clear. Nonetheless, this was simply meant to compare to what others have considered as notions of stationarity and not all together important for our results.
>
> 3a. We are not sure what you mean that we define the constrained min-max problem but study the unconstrained problem.  We would like to refer you back to the first paragraph of the paper (see line 19 where we define the sets X and Y to be Euclidean space) which defines the class of games we study to be on unconstrained spaces and also the beginning of Section 3.
>
> 3b. f* was defined in line 379 primarily to point out that we assume that the minimum function value is bounded for the result. In addition on unconstrained spaces, having f be bounded ensures existence of local minmax equilibrium.
>
>
> 4. We believe that this work stands on its own without empirical results, given that it is already nearly 50 pages with a significant amount of theoretical analysis. Moreover, we study a common algorithm which is already used in practice, so it is not entirely clear how empirical results would strengthen the paper. We note that a significant proportion of related papers on nonconvex-PL/SC zero-sum games do not contain empirical studies and also that often when they do the actual experiments reflect settings where the theoretical assumptions are violated.
>
> Finally, we would like to push back somewhat against your comment that the results are not very surprising. The results combine a number of nontrivial technical arguments and give convergence guarantees to game-theoretically meaningful equilibrium (specifically local minmax equilibria) in the most general class of games that has appeared in the literature to date. The fact that Danskin like results may hold in no way makes the results trivial (which we acknowledge you also agreed with), especially given that the inner problem is not actually being solved at each step (even approximately). Thus, we are showing that with the appropriate modifications (namely timescale separation), standard implementations of gradient descent ascent for these classes of problems have strong theoretical guarantees. Overall our belief is that despite the massive amount of work on these problem classes in the recent years, we fill a key missing gap by giving convergence to only game-theoretic equilibrium as opposed to first order notions of stationarity (which may in fact be local max-max points).  The significance of these results is akin to the results from nonconvex optimization that show almost sure asymptotic or finite-time convergence to local minimum, which themselves have been foundational to that field.
> Moreover, it is is not necessary for results to be surprising to be worthwhile or important.
>
> Thanks for your consideration of our response! Please let us know if you have any remaining concerns or if there is anything that would help you be able to recommend this paper as an accept.

---

> > ### Comment · Reviewer_gXSR · 2021-08-28
> > **Reviewer's Response**
> >
> > Thank you for the detailed response.
> >
> > Most of the concerns I had are clarified. I still believe that despite the meticulous and clean analysis presented in the paper, the results are not very surprising (especially when the gradient of the max value function is differentiable). That does not undervalue the importance of the results.
> >
> > I will be raising my score to 7.

---

### Author Response · Authors · 2021-08-24
**Looking for feedback from the reviewers: happy to clarify any remaining questions**

Thanks for your hard work reviewing. We just wanted to reach out and see if any of the reviewers had any comments back to our rebuttal? We are looking for feedback on whether the points made in the reviews have now been addressed. We are happy to answer any remaining questions regarding our rebuttal or the paper itself.

---

### Author Response · Authors · 2021-08-31
**Looking for feedback on rebuttal from Reviewer ZYYQ**

With the discussion period coming to a close soon, we wanted to reach out to reviewer ZYYQ and see if they have read our rebuttal and if it has cleared up the questions in the review and if there is anything else left that we need to answer.

---

### Public Comment · Authors · 2022-01-12
**Final Camera Ready Version**

We have posted the final camera ready version of the paper. We note that after the conference when reviewing our paper for the final camera ready deadline, we discovered an issue in one of the results. Specifically, at the end of the paper we had a result stating a finite-time convergence guarantee for $\tau$-GDA to a notion of approximate local minmax equilibrium in nonconvex-strongly-concave zero-sum games; however, we realized the proof contained some circular definitions of parameters. We have removed that particular result. This does not affect the main conclusion of the paper, which is that $\tau$-GDA has global convergence guarantees to strict local minmax equilibrium in nonconvex-PL and nonconvex-strongly-concave zero-sum games. Moreover, the other results we had of finite-time convergence guarantees to approximate stationary points also remain unchanged.

---

### Decision · Program_Chairs · 2021-09-27

**Decision:**

Accept (Poster)

**Comment:**

The authors study the behavior of simultaneous gradient descent-ascent with timescale separation (i.e. a slower learning rate for the descent dynamics compared to the ascent dynamics) to solve unconstrained and sequential min_x max_y f(x,y) optimization problems for objectives f(x,y) that are nonconvex wrt the variables of the outer (min) player and strongly concave or satisfy the PL condition wrt to the variables of the inner (y) player.

They focus on the stability analysis and convergence rates of GDA (and its perturbed gradient variant) to local minimax points. The definition of local min-max points is that the max player plays a global best response to min, while the min player chooses a local minimum for the function \max_{y \in B} f(.,y) where B is a ball of small radius. The authors show that local min-max points are stable/attracting for GDA dynamics, as long as the learning rate of the inner/max player is large enough compared to that of the min player. For nonconvex-strongly concave landscapes, they also show convergence in \tilde{O}(1/\eps^2) of perturbed GDA to an \esp- local min-max  and \tilde{O}(1/\eps^4) of perturbed SGDA.

Prior work had shown convergence to stationary points of \max_{y \in B} f(.,y) (for a slightly different algorithm) and this work does stability analysis to argue that the points it converges to are actually local minima. Stability analyses had also been done for related algorithms. So at a high level the results here are not surprising. Yet, they are not trivial. Doing the complete analysis to establish the results requires quite a bit of work, and this is the first paper that goes beyond stationarity. Given that min-max optimization for objectives that are not convex-concave is an interesting direction for the field, this is a solid contribution.